# Satellite validation strategy assessments based on the AROMAT campaigns

Alexis Merlaud[1], Livio Belegante[2], Daniel-Eduard Constantin[3], Mirjam Den Hoed[4], Andreas Carlos Meier[5], Marc Allaart[4], Magdalena Ardelean[6], Maxim Arseni[3], Tim Bösch[5], Hugues Brenot[1], Andreea Calcan[6], Emmanuel Dekemper[1], Sebastian Donner[8], Steffen Dörner[8], Mariana Carmelia Balanica Dragomir[3], Lucian Georgescu[3], Anca Nemuc[2], Doina Nicolae[2], Gaia Pinardi[1], Andreas Richter[5], Adrian Rosu[3], Thomas Ruhtz[7], Anja Schönhardt[5], Dirk Schuettemeyer[10], Reza Shaiganfar[8], Kerstin Stebel[9], Frederik Tack[1], Sorin Nicolae Vâjâiac[6], Jeni Vasilescu[2], Jurgen Vanhamel[1], Thomas Wagner[8], and Michel Van Roozendael[1]

[1]Royal Belgian Institute for Space Aeronomie (BIRA-IASB), Avenue Circulaire 3, 1180 Brussels, Belgium
[2]National Institute of R&D for Optoelectronics (INOE), Magurele, Street Atomistilor 409, Magurele 77125, Romania
[3]"Dunarea de Jos" University of Galati, Faculty of Sciences and Environment, Str. Domneasca, Nr. 111, Galati 800008, Romania
[4]Royal Netherlands Meteorological Institute (KNMI), De Bilt, The Netherlands
[5]Institute of Environmental Physics, University of Bremen (IUP-Bremen), Otto-Hahn-Allee 1, 28359 Bremen, Germany
[6]National Institute for Aerospace Research "Elie Carafoli" (INCAS), Bd. Iuliu Maniu no. 220, Bucharest, Romania
[7]Institute for Space Sciences, Free University of Berlin (FUB), Carl-Heinrich-Becker-Weg 6-10, 12165 Berlin, Germany
[8]Max-Planck-Institute for Chemistry (MPIC), Hahn-Meitner-Weg 1, 55128 Mainz, Germany
[9]Norwegian Institute for Air Research (NILU), Instituttveien 18, 2007 Kjeller, Norway
[10]European Space Agency (ESA-ESTEC), Keplerlaan 1, 2201 AZ Noordwijk, The Netherlands

**Correspondence:** A.Merlaud (alexism@oma.be)

**Abstract.**

The Airborne ROmanian Measurements of Aerosols and Trace gases (AROMAT) campaigns took place in Romania in September 2014 and August 2015. They focused on two sites: the Bucharest urban area and large power plants in the Jiu Valley. The main objectives of the campaigns were to test recently developed airborne observation systems dedicated to air

quality studies and to verify their applicability for the validation of spaceborne atmospheric missions such as the TROPOspheric Monitoring Instrument (TROPOMI)/Sentinel-5 Precursor (S5P). We present the AROMAT campaigns from the perspective of findings related to the validation of tropospheric $NO_2$, $SO_2$, and $H_2CO$. We also quantify the emissions of $NO_x$ and $SO_2$ at both measurement sites.

We show that tropospheric $NO_2$ vertical column density (VCD) measurements using airborne mapping instruments are

10 in principle well suited for satellite validation. The signal to noise ratio of the airborne $NO_2$ measurements is one order of magnitude higher than its spaceborne counterpart when the airborne measurements are averaged at the TROPOMI pixel scale. However, we show that the temporal variation of the $NO_2$ VCDs during a flight might be a significant source of comparison error. Considering the random error of the TROPOMI tropospheric $NO_2$ VCD ($\sigma$), the dynamic range of the $NO_2$ VCDs field extends from detection limit up to 37 $\sigma$ ($2.6 \times 10^{16}$ molec cm$^{-2}$) or 29 $\sigma$ ($2 \times 10^{16}$ molec cm$^{-2}$) for Bucharest and the

15 Jiu Valley, respectively. For both areas, we simulate validation exercises applied to the TROPOMI tropospheric $NO_2$ product.

These simulations indicate that a comparison error budget matching closely the TROPOMI optimal target accuracy of 25% can be obtained by when adding $NO_2$ and aerosol profile information to the airborne mapping observations, which constrains the investigated accuracy to within 28%. In addition to $NO_2$, our study also addresses the measurements of $SO_2$ emissions from power plants in the Jiu Valley, as well as a urban hotspot of $H_2CO$ in the center of Bucharest. For these two species, we conclude that the best validation strategy would consist in deploying ground-based measurement systems at well identified locations.

## 1  Introduction

Since the launch of the Global Ozone Monitoring Experiment (GOME, Burrows et al. (1999)) in 1995, spaceborne observations of reactive gases in the UV-visible range have tremendously improved our understanding of tropospheric chemistry. GOME mapped the large urban sources of $NO_2$ in North America and Europe, the $SO_2$ emissions from volcanoes and coal-fired power plants (Eisinger and Burrows, 1998), and the global distribution of $H_2CO$ with its maxima above East Asia and the tropical forests (De Smedt et al., 2008). Subsequent air-quality satellite missions expanded on the observation capabilities of GOME. Table 1 lists the past, present, and near-future nadir-looking satellite instruments dedicated to ozone and air quality monitoring with their sampling characteristics in space and time. The pixel size at nadir has shrunk from 320x40 $km^2$ (GOME) to 3.5x5.5 $km^2$ (TROPOMI, Veefkind et al. (2012), the original TROPOMI resolution of 7x5.5 $km^2$ was increased on 6 August 2019, MPC (2019)). This high horizontal resolution enables for instance to disentangle contradictory trends in ship and continental emissions of $NO_2$ in Europe (Boersma et al., 2015) or to distinguish the different $NO_2$ sources in oil sand mines in Canada (Griffin et al., 2019). The satellite-derived air quality products are now reliable enough to improve the bottom-up emission inventories (e.g. Kim et al. (2009), Fioletov et al. (2017), Bauwens et al. (2016)) and to be used in operational services, for instance to assist air traffic control with the near-real time detection of volcanic eruptions (Brenot et al., 2014). The bottom lines of Table 1 present the near-future perspective in spaceborne observation of the troposphere: a constellation of geostationary satellites will provide hourly observations of the troposphere above east Asia (GEMS, (Kim, 2012)), North America (TEMPO, Chance et al. (2013)), and Europe (Sentinel-4, Ingmann et al. (2012)). These new developments will open-up new perspectives for atmospheric research and air quality policies (Judd et al., 2018).

Validation is a key aspect of any spaceborne Earth observation mission. This aspect becomes even more important as the science matures and leads to more operational and quantitative applications. Validation involves a statistical analysis of the differences between measurements to be validated and reference measurements, which are independent data with known uncertainties (von Clarmann, 2006; Richter et al., 2014). The aim of validation is to verify that the satellite data products meet their requirements in terms of accuracy and precision. Table 2 presents such requirements for the TROPOMI-derived tropospheric vertical column densities (VCDs) of $NO_2$, $SO_2$, and $H_2CO$ (ESA, 2014). Richter et al. (2014) have discussed the challenges associated with the validation of tropospheric reactive gases. These challenges arise from the large variability in space and time of short-lived reactive gases, the dependency of the satellite products on different geophysical parameters (surface albedo, profile of trace gases and aerosols), the differences in vertical sensitivity between satellite and reference (ground-based or

airborne) measurements, and the small signals. An ideal validation study would involve a reference dataset of VCDs whose
well-characterized uncertainties would be small compared to those required for the investigated products. This reference dataset would cover a large amount of satellite pixels with adequate spatial and temporal representativeness at different seasons, places, and pollution levels. Beside the VCDs, the ideal validation exercise would also quantify the geophysical parameters that impact the retrieval of the investigated satellite products. In the real world however, Richter et al. (2014) points out that "the typical validation measurement falls short in one or even many of these aspects".

The first validations of the tropospheric $NO_2$ and $H_2CO$ VCD products of GOME involved in-situ samplings from aircraft (Heland et al., 2002; Martin et al., 2004). Such measurements may cover good fractions of satellite pixels but they miss the lower part of the boundary layer, where the trace gas concentrations often peak. Schaub et al. (2006) and Boersma et al. (2011) summarize other early validation studies for the tropospheric $NO_2$ VCDs retrieved from GOME, SCIAMACHY, and OMI. Several of these studies make use of the $NO_2$ surface concentration datasets from air quality monitoring networks. Compared
to campaign-based data acquisition, operational in-situ networks provide long-term measurements, but their comparison with satellite products relies upon assumptions on the $NO_2$ profile. Other validation studies use remote-sensing from the ground and aircraft, in particular based on the Differential Optical Absorption Spectroscopy (DOAS) technique (Platt and Stutz, 2008), which is also the basis for the retrieval algorithms of the satellite-derived products. In comparison with in-situ measurements, DOAS has the benefit of being directly sensitive to the column density of a trace gas, i.e. the same geophysical quantity as
the one retrieved from space. Heue et al. (2005) conducted the first comparison between a satellite-derived product (SCIA-MACHY tropospheric $NO_2$) and airborne DOAS data. Many validation studies also use ground-based DOAS measurements, in particular since the development of the Multi-AXis DOAS (MAX-DOAS) technique (Hönninger et al., 2004). MAX-DOAS measurements are valuable for validation due to their ability to measure integrated columns at spatial scales comparable to the satellite ground pixel size. Moreover, they broaden the scope of validation activities since they also provide limited profile
information on both trace gases and aerosols (Irie et al., 2008; Brinksma et al., 2008; Ma et al., 2013; Kanaya et al., 2014; Wang et al., 2017; Drosoglou et al., 2018). The limitations of using the MAX-DOAS technique for validation arise from their still imperfect spatial representativeness compared to typical satellite footprints and to some extent from their limited sensitivity in the free troposphere. Spatial representativeness has often been invoked to explain the apparent low bias of the OMI tropospheric $NO_2$ VCDs in urban conditions (Boersma et al., 2018).

The unprecedented horizontal resolution enabled by the last generation of air-quality space-based instruments motivated preparatory field studies around polluted areas in North America (DISCOVER-AQ, https://discover-aq.larc.nasa.gov), Europe (AROMAT and AROMAPEX, Tack et al.,2019) and Korea (KORUS-AQ, https://www-air.larc.nasa.gov/missions/korus-aq/). These campaign activities quantified key pollutants ($NO_2$, $SO_2$, $O_3$, $H_2CO$, and aerosols) and assessed practical observation capabilities of future satellite instruments while preparing for their validation. They combined ground-based and airborne
measurements. DISCOVER-AQ involved the deployment of the Geostationary Trace gas and Aerosol Sensor Optimization instrument (GEOTASO, Leitch et al.,2014; Nowlan et al.,2016) and of the Geostationary Coastal and Air Pollution Events (GEO-CAPE) Airborne Simulator (GCAS, Kowalewski and Janz,2014;Nowlan et al.,2018). In Europe, the two AROMAT campaigns, which took place in Romania in September 2014 and August 2015, demonstrated a suite of new instruments such

as the Airborne imaging DOAS instrument for Measurements of Atmospheric Pollution (AirMAP, Schönhardt et al.,2015; Meier et al.,2017), the $NO_2$ sonde (Sluis et al., 2010), and the Small Whiskbroom Imager for atmospheric compositioN monitoring (SWING, Merlaud et al.,2018). Different airborne imagers were intercompared and further characterized during the AROMAPEX campaign in April 2016 (Tack et al., 2019).

Two aforementioned publications focused on the AirMAP and SWING operations during the 2014 AROMAT campaign (Meier et al., 2017; Merlaud et al., 2018). In this work, we present the overall instrumental deployment during the two campaigns and analyze the relevance of these measurements for the validation of several air quality satellite products: tropospheric $NO_2$, $SO_2$ and $H_2CO$ VCDs. The datasets collected during AROMAT fulfill several requirements of the ideal validation study, as described above. We further investigate the strengths and limitations of the acquired data sets.

The paper is structured as follows: Section 2 describes the two target areas and the deployment strategy. Section 3 characterizes the investigated trace gases fields in the sampled areas. Section 4 presents a critical analysis of the strengths and limitations of the campaign results while elaborating on recommendations for future validation campaigns in Romania. Eventually, we use the AROMAT measurements to derive $NO_x$ and $SO_2$ fluxes from the two sites. The Supplement presents technical details on the instruments operated during the campaigns and presents additional information and measurements.

## 2 Target areas and deployment strategy

This section presents the two target areas of the AROMAT campaigns, Bucharest and the Jiu Valley. It also lists available studies on air quality at these two sites as well as logistical aspects of relevance.

Figure 1 presents a map of the tropospheric $NO_2$ vertical column densities (VCDs) above Romania, derived from OMI measurements (Levelt et al., 2006) and averaged between 2012 and 2016. The map also indicates the position of the 8 largest cities of the country. Compared to highly polluted areas in western Europe such as northern Belgium or the Netherlands, Romania appears relatively clean at the spatial resolution of the satellite data. There are however two major $NO_2$ sources clearly visible from space, which appear to be of similar magnitude with $NO_2$ columns around 2.5 x $10^{15}$ molec cm$^{-2}$: the Bucharest area and the Jiu Valley, northwest of Craiova. For the latter, the $NO_2$ enhancement is due to a series of large coal-fired thermal power plants.

### 2.1 Bucharest

Bucharest (44.4° N, 26.1° E) is the capital and largest city (1.9 million inhabitants according to the 2011 census) of Romania. Within its administrative borders, the city covers an area of 228 km$^2$. Adding the surrounding Ilfov county, the total Bucharest metropolitan area numbers 2.3 million inhabitants in 1.583 km$^2$. The built-up areas are mainly located within a ring road whose diameter is around 20 km.

Iorga et al. (2015) described in detail the Bucharest Greater Area in the context of an extensive study of the air quality in the city between 2005 and 2010. Bucharest is located in a low-altitude plain, with a maximum altitude of 92 m a.s.l. The geographic configuration of the Carpathian Mountains explains the dominant northeast winds.

The $NO_2$ VCDs seen from space above Bucharest appear lower than over western European sites at the resolution of OMI (see Fig. S1 in the Supplement). However, this is partly due to the dilution effect for this relatively small and isolated source. Local studies based on the 8 air quality stations inside the city point out that, regarding local PM and $NO_x$ levels, Bucharest is amongst the most polluted cities in Europe (Alpopi and Colesca, 2010; Iorga et al., 2015). The city center is the most heavily polluted, with concentrations of pollutants well above the European thresholds. For instance, the annual mean concentration of $NO_2$ at the traffic stations was about 57 $\mu$ g.m$^{-3}$ in 2017 (EEA, 2019), when the EU limit is 40 $\mu$ g.m$^{-3}$. Stefan et al. (2013) have shown the importance of local conditions and anthropogenic factors in air quality analysis in areas close to Bucharest, during two weeks of measurements in 2012. Iorga et al. (2015) and Grigoraş et al. (2016) showed that the main $NO_x$ contributions came from traffic and production of electricity, spread over about 10 medium-size thermal power plants within the city.

Figure 2 shows the Bucharest metropolitan area and the flight tracks of the two scientific aircraft used during AROMAT-2 (the FUB Cessna-207 and the INCAS BN-2). Note that the BN-2 tracks are actually a good indication of the Bucharest ring road. We were not allowed to cross the ring road with the BN-2, except in the North of the city. The figure also pinpoints important locations for the AROMAT campaigns. The FUB Cessna took-off and landed at the Baneasa international airport, located 8 km north of Bucharest city center (44.502° N, 26.101° E). The INCAS BN-2 also used Baneasa airport during AROMAT-2, but this plane was mainly based at the Strejnicu airfield (44.924° N, 25.964° E), which lies 60 km north of Bucharest, near Ploiesti. The UAV operations in Bucharest during AROMAT-1 were performed at Clinceni airfield (44.359° N, 25.931 ° E). The latter is located in the southwest of the city, 7 km west of the INOE observatory in Magurele (44.348°N, 26.031°E).

## 2.2 The Jiu Valley between Targu Jiu and Craiova

The second $NO_2$ plume in Fig. 1 lies around 250 km west of Bucharest. It corresponds to a series of four thermal power plants located along the Jiu river between the cities of Targu Jiu (82,000 inhabitants, 45.03°N, 23.27°E) and Craiova (269,000 inhabitants, 44.31°N, 23.8°E). These plants were built in this area due to the presence of lignite (brown coal), which is burned to produce electricity.

The altitude of the valley ranges from 268 m a.s.l. in Targu Jiu to 90 m in Craiova. The valley is surrounded by moderately elevated hills (400 m a.s.l.). Due to the orography, the prevailing wind directions is from southwest to southeast.

Beside $NO_2$, the $SO_2$ emissions from these plants are also visible from space, as first reported by Eisinger and Burrows (1998) using GOME data. Since 2011, the OMI-derived trends above the area indicate that the emissions of $SO_2$ have been decreasing, while those of $NO_2$ are stable (Krotkov et al., 2016). This is related to the installation of flue gas desulfurization (FGD) systems, which was part of environmental regulations imposed on Romania following its entry in the European Union in 2007.

Figure 3 presents a map of the Jiu Valley area with the four power plants. The map also shows the tracks of the two airborne platforms (the FUB Cessna and an ultralight operated by UGAL) operated in this area during AROMAT-2. Table S1 in the

Supplement presents the geographical positions, nominal capacities, and smokestack heights of the four power plants. From north to south, the plants are named according to their locations: Rovinari, Turceni, Isalnita and Craiova II.

During the AROMAT campaigns, we focused in particular on the emissions of the Turceni power plant (44.67°N, 23.41°E). With a nominal capacity of 1650 MW, it is the largest electricity producer in Romania. The Turceni power plant is located in a rural area, 2 km ESE of the village of Turceni. The plant emits aerosols, NOx, and $SO_2$ from the 280 m high smokestacks.

Scientific studies on air quality inside the Jiu Valley are sparse. Previous measurements performed by INOE during a campaign in Rovinari in 2010 indicated elevated volume mixing ratios of $NO_2$ (up to 30 ppb) and of $SO_2$ (up to 213 ppb) (Nisulescu et al., 2011; Marmureanu et al., 2013). The maximum ground concentrations occurred in the morning, before the planetary boundary layer development. Mobile-DOAS observations performed in 2013 revealed columns of $NO_2$ up to 1 x $10^{17}$ molec $cm^{-2}$ (Constantin et al., 2015).

## 2.3 Groups, instruments, and platforms

The AROMAT consortium consisted of research teams from Belgium (BIRA-IASB), Germany (IUP-Bremen, FUB, MPIC), The Netherlands (KNMI), Romania (University "Dunarea de Jos" of Galati, hereafter UGAL, National Institute of R&D for Optoelectronics, hereafter INOE, and National Institute for Aerospace Research "Elie Carafoli", hereafter INCAS), and Norway (NILU). The AROMAT consortium had a common focus on measuring the tropospheric composition using various techniques.

Figure 4 illustrates the typical instrumental deployment during the campaigns. The set-up combined airborne and ground-based measurements to sample the 3-D chemical state of the lower troposphere above polluted areas. The Supplement presents the main atmospheric instruments operated during the two campaigns, classified into airborne, ground-based, remote sensing, and in-situ sensors. The primary target species during AROMAT-1 were $NO_2$ and aerosols while the observation capacities expanded in AROMAT-2 through the improvements of the AirMAP and SWING sensors for $SO_2$ measurements and the deployments of other instruments such as $SO_2$ cameras, DOAS instruments targeted to $H_2CO$, and a PICARRO instrument to measure water vapor, methane, CO, and $CO_2$.

We used two small tropospheric aircraft: the Cessna-207 from FUB, and the Britten-Norman Islander (BN-2) from INCAS. The Cessna was dedicated to remote sensing. It mainly performed mapping flights at 3 km a.s.l. for the airborne imagers, while parts of the ascents and descents were used to measure aerosol extinction profiles with the FUBISS-ASA2 instrument. The BN-2, which was only used during AROMAT-2, was dedicated to in-situ measurements around Bucharest between surface and 3000 m a.s.l.. In AROMAT-2, there was also an ultralight aircraft used by UGAL for nadir-DOAS observations in the Jiu Valley. The ultralight aircraft typically flew between 600 and 1800 m a.s.l. Two UAVs, operated by INCAS and UGAL flew during AROMAT-1. These measurements were not repeated during AROMAT-2 since the coverage of the UAVs was too limited, both in horizontal and vertical direction. Finally, we also launched balloons carrying $NO_2$ sondes from Turceni and performed Mobile-DOAS measurements from several cars during both campaigns. The Supplement provides more details about the practical deployments during the campaigns.

## 2.4 The 2014 AROMAT campaign

The AROMAT-1 campaign took place between 1 and 13 September 2014. The operations started in Bucharest with the continuous observations from the Romanian Atmospheric 3D Observatory (RADO, Nicolae et al. (2010)) in Magurele and synchronized car-based Mobile-DOAS observations around the Bucharest ring road and within the city. During the first two days of the campaign, the INCAS UAV flew from the Clinceni airfield with two different aerosol payloads (the TSI Dust Trak DRX and TSI aerosol particle sizer) up to an altitude of 1.2 km a.s.l. The Cessna was not allowed to fly over the city but performed loops above the ring road at a low altitude of 500 m a.s.l. The remote sensing measurements stopped on 4 September due to bad weather. On 5 and 6 September, we collected data only from the ground, and in broken cloud conditions.

On 7 September 2014, part of the campaign crew moved to the Jiu Valley. We installed the INOE mobile laboratory (in-situ monitors, MILI lidar, and ACSM) in Turceni and performed the first UAV flights around the power plant on 8 September 2014 with the $NO_2$ sonde and SWING. On the same day in Bucharest, the Cessna flew above the city with AirMAP and Mobile-DOAS operated on the ground. On the following day, 9 September 2014, the Cessna did a second mapping of Bucharest and we started to launch balloons from Turceni, carrying the $NO_2$ sonde. In total, 11 balloons were launched between 8 and 12 September 2014, out of which 10 led to successful measurements. Technical issues with both the UAV and the Cessna interrupted the flights for a couple of days. The UAV operations started again with a SWING flight on 10 September 2014. On 11 September 2014, the AirMAP and SWING flew in coincidence above Turceni, on the Cessna and the UAV respectively, and we performed two more short SWING-UAV flights. On 12 and 13 September, we performed two more Cessna flights above the Jiu Valley but the weather conditions were degrading. During the entire second week of the campaign, Mobile-DOAS measurements were performed in Turceni and around the other power plants of the Jiu Valley.

Table S5 in the Supplement summarizes the main measurement days during AROMAT-1, specifying if the measurements were taken in Bucharest or in the Jiu Valley. The "golden days" of the AROMAT-1 campaigns are 2, 8, and 11 September 2014. These days are particularly interesting due to good weather conditions and coincident measurements. On 2 September 2014, we operated the three Mobile-DOAS together around Bucharest. On 8 September 2014, we flew AirMAP above Bucharest with the UGAL and MPIC Mobile-DOAS on the ground. Finally, on 11 September 2014, SWING and AirMAP were time-coincident above the Turceni power plant, and two balloons sampled the vertical distribution of $NO_2$.

## 2.5 The 2015 AROMAT-2 campaign

The AROMAT-2 campaign took place between 17 and 31 August 2015. We started in Bucharest with car-based Mobile-DOAS measurements and observations at RADO. The INOE mobile lab was installed in Turceni on 19 August 2015, followed by an $SO_2$ camera (instrument described in Kern et al., 2015; Stebel et al., 2015) and $NO_2$ camera (Dekemper et al., 2016). Poor weather conditions limited the relevance of the measurements during the first days of the campaign. Two Mobile-DOAS teams in Bucharest moved from Bucharest to the Jiu Valley on 23 August 2015. From then, the weather was fine until the end of the campaigns, and valuable data were collected during all days between 24 and 31 August 2015.

In the Jiu Valley, the crew was based in Turceni and most of the static instruments were installed at a soccer field. Beside the INOE mobile lab with in-situ samplers, the scanning lidar, $SO_2$ cameras and the $NO_2$ camera pointed to the power plant plume. The $NO_2$ camera acquired images until 25 August 2015. The car-based Mobile-DOAS operated in the Valley between the different power plants. From 24 August, the $SO_2$ cameras were split: one of them stayed in the soccer field, the two others were installed at several points around Turceni. Also on 24 August, the UGAL ultralight took off from Craiova and flew to the Jiu Valley until Rovinari, carrying the ULM-DOAS instrument. This experiment was repeated on 25, 26, and 27 August. On 28 August 2015, the Cessna flew above Turceni with AirMAP and SWING.

In Bucharest, the BN-2 flew first on 25 August 2015. It took off from Strejnicu and carried various in-situ instruments: the TSI nephelometer and Aerosol Particle Sizer, the $NO_2$ CAPS, the PICARRO, and the KNMI $NO_2$ sonde, and flew in a loop pattern at 500 m a.s.l. around the city ring road. After this test flight, the aircraft performed 6 flights between 27 and 31 August 2015, which included soundings around Baneasa and Magurele, up to 3300 m a.s.l. On 30 and 31 August 2015, the Cessna mapped the city of Bucharest, performing two flights per day. It also performed soundings to measure AOD profiles with the FUBISS-ASA2 instrument (Zieger et al., 2007).

Table S6 in the Supplement summarizes the measurements of the AROMAT-2 campaign, specifying if the measurements were taken in Bucharest or in the Jiu Valley. Compared to the AROMAT-1 campaign, a larger number of instruments took part and also a larger number of 'golden days' occurred. All the days between 24 and 31 August 2015 led to interesting measurements. Regarding intercomparison exercises for the airborne imagers, the best days are 28 August 2015 (Jiu Valley) and 31 August 2015 (Bucharest).

## 3 Geophysical results

This section presents selected findings related to tropospheric $NO_2$, $SO_2$ and $H_2CO$ in the two target areas. The Supplement gives details about the instruments involved in these observations and presents additional measurements in Bucharest and the Jiu Valley.

### 3.1 Bucharest

#### 3.1.1 Horizontal distribution of $NO_2$

Figure 5 presents two maps of the AROMAT $NO_2$ measurements performed with the AirMAP, CAPS, and MPIC mobile-DOAS instruments above Bucharest, on 30 (Sunday afternoon, left panel) and 31 (Monday afternoon, right panel) August 2015. AirMAP is a remote sensing instrument that mapped the $NO_2$ VCDs from the Cessna at 3 km a.s.l. and produced the continuous map. The CAPS is an in-situ instrument, it was operated on the BN-2 and sampled the air at 300 m a.s.l. and performed vertical soundings above Magurele. The MPIC Mobile-DOAS mainly drove along the Bucharest ring road.

The datasets of Fig. 5 reveal large differences of $NO_2$ amounts on Sunday 30 August 2015 compared to Monday 31 August 2015. On Sunday afternoon, the $NO_2$ VCDs peak around 1.5 x $10^{16}$ molec $cm^{-2}$. On Monday, the $NO_2$ plume spread from the

center to the northeast of the city. The observed $NO_2$ VCDs were smaller than the detection limit upwind and reach up to 3.5 x $10^{16}$ molec cm$^{-2}$ inside the plume. The $NO_2$ VMR measured with the CAPS was close to the detection limit on Sunday while it reached 5 ppb inside the plume on Monday 31 August 2015. Note that the time difference between both measurements partly explain the systematic differences between AirMAP and the MPIC Mobile-DOAS at the eastern part of the ring road on 31 August 2015. Meier (2018) compared the two instruments during the morning flight (between 07:00 and 09:30 UTC), which includes more simultaneous observations. Considering only collocated measurements with a maximum time difference of 45 minutes, the comparison reveals a good agreement when averaging the forward and backward-looking Mobile-DOAS $NO_2$ VCDs. The MPIC/AirMAP slope is 0.93 while the correlation coefficient of 0.94. The remaining discrepancy may be explained by AMFs errors and differences in time and horizontal sensitivity. Figure S2 in the Supplement presents this quantitative comparison.

Figure 6 presents collocated CAPS and AirMAP $NO_2$ measurements on 31 August 2015. The BN-2 carrying the CAPS flew from Magurele to the East of Bucharest, remaining outside the city ring at 300 m a.s.l. between 12:30 and 12:55 UTC, while AirMAP onboard the Cessna was mapping the city between 12:00 and 13:30 UTC. We extracted the AirMAP $NO_2$ VCDs at the position of the CAPS observations. The figure confirms that the two instruments detected the plume at the same place. This suggests that along this portion of the flight, which was inside the plume but outside the city, the $NO_2$ VMR measured at 300 m a.s.l. may be used as a proxy for the $NO_2$ VCD. Indeed, the BLH was about 1500m (Fig.S8 in the Supplement and discussion therein) during these observations. Assuming a constant $NO_2$ VMR of 3.5 ppb in the boundary layer leads to a $NO_2$ VCD of 1.4 x $10^{16}$ molec cm$^{-2}$. This estimate is close to the AirMAP $NO_2$ VCD observed in the plume (Fig. 6). When measured at 300 m a.s.l., the $NO_2$ VMR thus seems a good estimate of its average within the boundary layer. Note that this finding is specific to the configuration in Bucharest where we flew at 10 km from the city center and does not apply to our measurements in the exhaust plume of the Turceni power plant (Fig. 9). Future campaigns should include vertical soundings inside the Bucharest plume to further investigate its $NO_2$ vertical distribution.

### 3.1.2 Horizontal distribution of $H_2CO$

Figure 7 shows the $H_2CO$ and $NO_2$ VCDs measurements from the IUP-Bremen nadir instrument operated onboard the Cessna on 31 August 2015 (morning flight), together with the MPIC Mobile-DOAS measurements. The airborne data shown correspond to the second overpass (07:46-08:23 UTC) while the Mobile-DOAS were recorded between 08:13 and 10:00 UTC. The $H_2CO$ VCDs range between $1\pm0.25$ x $10^{16}$ molec cm$^{-2}$ and $7.5\pm2$ x $10^{16}$ molec cm$^{-2}$, a maximum observed inside the city. We estimated the $H_2CO$ reference column for the airborne data using the Mobile-DOAS measurements. Both $NO_2$ and $H_2CO$ are in good agreement when comparing their distributions as seen from the airborne and ground-based instruments. However, if the highest $H_2CO$ VCDs are found above the Bucharest city center, they are not coincident with the $NO_2$ maximum, as can be seen comparing the upper and lower panels of Fig. 7, for instance on the second Cessna flight line from the north.

The $H_2CO$ hotspot observed above Bucharest is mainly anthropogenic. Indeed, biogenic emissions typically account for 1 to 2 x $10^{16}$ molec cm$^{-2}$ (J.-F. Müller, personal communication), in agreement with the background VCDs measured by the Mobile-DOAS along the Bucharest ring. During the measurements, the wind was blowing from south and west. The difference

between $NO_2$ and $H_2CO$ spatial patterns may be explained by the different origins of $NO_x$ compared to $H_2CO$ or by the formation time of $H_2CO$ through the oxidation of VOCs.

Anthropogenic hotspots of $H_2CO$ have already been observed, e.g. above Houston (Texas), an urban area which includes significant emissions from transport and petrochemical industry (Parrish et al., 2012; Nowlan et al., 2018). Nowlan et al. also deployed an airborne DOAS nadir instrument, they reported $H_2CO$ VCDs up to 5 x $10^{16}$ molec $cm^{-2}$ in September 2013.

## 3.2   The Jiu Valley

### 3.2.1   Spatial distribution of $NO_2$

Figure 8 presents the horizontal distribution of the $NO_2$ VCDs in the Jiu Valley measured with the MPIC Mobile-DOAS on 23 August 2015 between 08:07 and 14:16 UTC. The figure shows elevated $NO_2$ VCDs close to the four power plants listed in Table S1 of the Supplement, with up to $8x10^{16}$ molec $cm^{-2}$ downwind of Turceni and Rovinari. In comparison, the area East of Craiova is very clean, with typical $NO_2$ VCDs under $1x10^{15}$ molec $cm^{-2}$.

    The situation of Fig. 8 is characteristic of the conditions encountered in the Jiu Valley, with high $NO_2$ VCDs observed
north and west of the plants due to the prevailing wind directions. During both campaigns, we observed maximum $NO_2$ VCDs reaching up to $1.3x10^{17}$ molec $cm^{-2}$ close to the plants with Mobile-DOAS instruments.

    Figure 10 (upper panels) shows the AirMAP and SWING $NO_2$ VCDs measured around the Turceni power plant on 28 August 2015. The two airborne instruments largely agree, detecting $NO_2$ VCDs up to $8x10^{16}$ molec $cm^{-2}$ in the exhaust plume of the power plant. Figure S5 in the Supplement (upper panel) extracts the AirMAP and SWING $NO_2$ VCDs along the
path of the ground-based BIRA Mobile-DOAS measurements and compares the three datasets. The airborne data correspond to three portions of flight lines recorded between 09:54 and 10:17 UTC. The BIRA Mobile-DOAS instrument was sampling the plume during this time so the maximum time difference is 23 minutes. This comparison confirms the good agreement for the airborne instruments but indicates that comparing airborne nadir-looking DOAS with ground-based zenith Mobile-DOAS instruments is not straightforward in these conditions. Table S2 in the Supplement gives the typical AMFs used in this analysis
for airborne and zenith-only Mobile-DOAS. When observed with the Mobile-DOAS, the plume shows higher $NO_2$ VCDs and appears narrower than with the airborne instruments. This is partly related to air mass factor uncertainties, but they can not explain alone such a discrepancy. Close to the power plant, the plume is very thin and heterogeneous which leads to 3-D effects in the radiative transfer, as suggested in a previous AROMAT study (Merlaud et al., 2018). In these conditions, the 1-D atmosphere of the radiative transfer models used to calculate the airborne AMFs may not be realistic enough and bias the
VCDs measured from the aircraft.

    Figure 9 shows those AROMAT-1 $NO_2$ sonde measurements above Turceni which detected the plume. The $NO_2$ is not well-mixed in the boundary layer, with maxima aloft and lower VMRs close to the surface. This is understandable so close to the source, as high-temperature $NO_x$ is emitted from the 280 m high stack. In these balloon-borne datasets, the observed maximum $NO_2$ VMR is about 60 ppb inside the plume, and the $NO_2$ VMR vanishes above 1200 m a.s.l.. These results suggest

that airborne measurements with the ULM-DOAS, which can fly safely at 1500 m a.s.l., can provide reliable measurements of the integrated column amount inside the plume.

### 3.2.2 Horizontal distribution of $SO_2$

Figure 10 (lower panels) presents the $SO_2$ horizontal distributions measured around Turceni with AirMAP (lower left panel) and SWING (lower right panel) on 28 August 2015. The maps show the plume from the Turceni plant transported in the northwest direction, and other areas with elevated $SO_2$ VCDs in the east and south of Turceni. Meier (2018) presents in detail these AirMAP $SO_2$ observations and compares them with SWING results. Figure S4 in the Supplement shows the corresponding time series of SWING and AirMAP $SO_2$ DSCDs. It is found that the AirMAP-derived $SO_2$ columns inside the plume $SO_2$ reach $6x10^{17}$ molec $cm^{-2}$ and that the AirMAP and SWING $SO_2$ VCDs agree within 10%. Moreover, for these airborne data, the $SO_2$ horizontal distribution broadly follows that of $NO_2$. The discrepancies can be explained by the different lifetimes of the two species.

As for $NO_2$, it appears difficult to quantitatively relate the airborne and Mobile-DOAS $SO_2$ VCDs observations in the close vicinity of the power plant. As shown in Fig. S5 of the Supplement (lower panel), the maximum $SO_2$ VCD measured from the ground on the road close to the factory amounts to $1.3 \times 10^{18}$ molec $cm^{-2}$ while from the aircraft, the $SO_2$ VCD reached $8 \times 10^{17}$ molec $cm^{-2}$. Part of this difference can be explained by 3-D effects on the radiative transfer, as for $NO_2$. As discussed below, it seems easier to compare the $SO_2$ flux.

## 4 Discussion

In this section, we develop the lessons learned from our study for the validation of satellite observations of the three investigated tropospheric trace gases, namely $NO_2$, $SO_2$, and $H_2CO$. For each molecule, we discuss the benefit of conducting such airborne campaigns as well as the choice of Romania as a campaign site. In the last part of the section, we also estimate the $NO_x$ and $SO_2$ emissions from Bucharest and from the power plants of the Jiu Valley, using the different datasets of the campaigns.

### 4.1 Lessons learned for the validation of space-borne $NO_2$ VCDs

#### 4.1.1 Number of possible pixels and dynamic range at the TROPOMI resolution

Regarding Bucharest, the mapped area of Fig. 5 (right panel) virtually covers 43 TROPOMI near-nadir pixels. Averaging the high spatial resolution AirMAP $NO_2$ VCDs within these 43 hypothetical TROPOMI measurements reduces the dynamic range of the observed $NO_2$ field. The latter decreases from $3.5x10^{16}$ to $2.6x10^{16}$ molec $cm^{-2}$ (37 $\sigma$ where $\sigma$ is the required precision on the tropospheric $NO_2$ VCD). Nevertheless, 33 of the 43 hypothetical TROPOMI pixels exhibits a $NO_2$ VCD above the required 2-$\sigma$ random error for TROPOMI ($1.4x10^{15}$ molec $cm^{-2}$).

Regarding the Jiu Valley, a similar exercise based on our measurments on 28 August 2012 (Fig. 10, upper panel) leads to 48 near-nadir TROPOMI pixels, out of which 35 would have a $NO_2$ VCD above the 2-$\sigma$ TROPOMI error. The largest $NO_2$ tropospheric VCD seen by TROPOMI would be around $2 \times 10^{16}$ molec cm$^{-2}$ (29 $\sigma$ for TROPOMI).

### 4.1.2   Characterization of the reference measurements

Table 3 summarizes the $NO_2$ observations during the AROMAT campaigns. For each instrument, the table indicates the measured range of $NO_2$ VCDs (or VMRs), the ground sampling distance and a typical detection limit and bias. Regarding DOAS instruments, we estimated the detection limits on the $NO_2$ VCDs from typical 1-$\sigma$ DOAS fit uncertainties divided by typical air mass factors (AMF). Table S2 in the Supplement presents these typical AMFs and detection limits. The 1-$\sigma$ DOAS fit uncertainty is instrument specific and an output of the DOAS fitting algorithms. The AMF depends on the observation's geometry, atmospheric and surface optical properties. Uncertainties on the AMF usually dominate the systematic part of the error for the DOAS measurements. Therefore, for these instruments, the bias given in Table 3 corresponds to the uncertainty in their associated AMF.

Combined with the ground sampling distance, the detection limit enables one to quantify the random uncertainty of a reference observation at the satellite horizontal resolution. Indeed, considering reference measurements averaged within a satellite pixel, the random error associated with the averaged reference measurements decreases with the square root of the number of measurements, following Poisson statistics. For instance, a continuous mapping performed with SWING at a spatial resolution of 300 x 300 m$^2$ inside a TROPOMI pixel of 3.5 x 5.5 km$^2$ would lead to 214 SWING pixels. Averaging the $NO_2$ VCDs of these SWING pixels would divide the SWING original uncertainty ($1.2 \times 10^{15}$ molec.cm$^{-2}$) by $\sqrt{214}$, leading to $8.2 \times 10^{13}$ molec.cm$^{-2}$, about one tenth of the random error of TROPOMI ($7 \times 10^{14}$ molec.cm$^{-2}$) given in table 2.

However, the temporal variation of the $NO_2$ VCDs further adds uncertainty to the reference measurements when comparing them with satellite data. The validation areas typically extend over a few tens of kilometers. At this scale, satellite observations are a snapshot in time of the atmospheric state, while an airborne mapping typically takes one or two hours.

Figure 11 illustrates our estimation of the temporal variation of the $NO_2$ VCDs comparing consecutive AirMAP overpasses above Bucharest from the morning flight of 31 August 2015. During this flight, the Cessna covered the same area three times in a row between 07:06 and 08:52 UTC. Figure S3 in the Supplement presents the corresponding AirMAP and SWING $NO_2$ DSCDs. For each AirMAP overpass, we averaged the $NO_2$ VCDs at the horizontal resolution of TROPOMI (see previous section). The standard deviation of the differences between two averaged overpasses then indicates the random part of the $NO_2$ VCDs temporal variation during an aircraft overpass. This standard deviation is $3.7 \times 10^{15}$ and $4.2 \times 10^{15}$ molec cm$^{-2}$, respectively between the first and second, and second and third overpass. Hereafter, we used $4 \times 10^{15}$ molec cm$^{-2}$ as random error due to the temporal variation.

Clearly, the $NO_2$ VCD temporal variation depends on characteristics of a given validation experiments, such as the source locations and the wind conditions during the measurements. The temporal variation also depends on the time of the day and we base our estimate here on measurements around 11:00 LT while TROPOMI overpass is at 13:30 LT. In the studied case however, this error source is larger for the reference measurements than the TROPOMI precision ($7 \times 10^{14}$ molec cm$^{-2}$). This is

quite different from using static MAX-DOAS as reference. The latter are usually averaged within one hour around the satellite overpass. Compernolle et al. (2020) quantify the temporal error for MAX-DOAS $NO_2$ VCDs, typically ranging between 1 to $5x10^{14}$ molec $cm^{-2}$. In the next section, we investigate the effect of underestimating the temporal random error.

### 4.1.3  Simulations of validation exercises in different scenarios

We simulated TROPOMI Cal/Val exercises with the spatially averaged AirMAP observations described in Sect. 4.1.1. We considered these averaged AirMAP $NO_2$ VCDs as the ground truth in simulated TROPOMI pixels, on which we added Gaussian noise to build synthetic satellite and reference $NO_2$ VCDs datasets. For the synthetic satellite observations, the noise standard deviation corresponded to the TROPOMI random error (the precision in Table 2). For the synthetic airborne observations, we added in quadrature the aforementioned averaged airborne shot noise (e.g. $7x10^{13}$ molec $cm^{-2}$ for SWING) and temporal error ($4x10^{15}$ molec $cm^{-2}$, which we assumed to be also realistic around Turceni). We then applied weighted orthogonal distance regressions to a series of such simulations to estimate the uncertainty on the regression slope. This led to slope uncertainties of about 6% and 10% in Bucharest and Turceni, respectively.

In a real-world validation experiment, this regression slope would quantify the combined biases of the two $NO_2$ VCDs datasets (satellite and reference). These biases mainly originate from errors in the AMFs, resulting in particular from uncertainties on the $NO_2$ and aerosol profiles, and on the surface albedo. To some extent, these quantities can be measured from an aircraft with the type of instrumentation deployed in the AROMAT activity. The ground albedo can be retrieved with the DOAS instruments by normalizing uncalibrated airborne radiances to a reference area with known albedo (Meier et al., 2017) or by using a radiometrically calibrated DOAS sensor (Tack et al., 2019). The $NO_2$ and aerosol profiles can be measured with in-situ instruments such as a CAPS $NO_2$ monitor and a nephelometer. For legal reasons, vertical soundings are difficult above cities. One can measure the $NO_2$ and aerosol profile further down in the exhaust plume, once the latter is above rural areas. The conditions inside the city can be different and this motivates the deployment of ground-based instruments, e.g. sunphotometers and MAX-DOAS, inside the city.

Regarding uncertainties on the references AMFs, the benefit of knowing the aerosol and $NO_2$ profile appears when comparing the AMF error budget for airborne measurements above Bucharest (26%, Meier et al. (2017)) and above the Turceni power plant (10%, Merlaud et al. (2018)). In the latter case, there was accurate information on the local $NO_2$ and aerosol profiles thanks to the lidar and the balloon-borne $NO_2$ sonde, respectively. We used these two AMF uncertainties to estimate a total possible bias between reference and satellite observations.

Table 6 presents total error budgets for different scenarios of validation exercises using reference airborne mapping to validate spaceborne tropospheric $NO_2$ VCDs. We estimated the random and systematic uncertainties between satellite and reference measurements with SWING and AirMAP, including (or not) profile information on the aerosols and $NO_2$ VMR, and for measurements over Bucharest or Turceni. Note that we considered 25% for the satellite accuracy. The temporal error of the airborne measurements clearly dominates the total random error, making the differences in detection limit between AirMAP and SWING irrelevant for this application. Adding the profile information on the other hand reduces the total multiplicative

bias from 37% to 28% or 29% in Bucharest and Turceni. This quantifies the capabilities of such airborne measurements for the validation of the imaging capabilities of TROPOMI regarding the $NO_2$ VCDs above Bucharest and the Jiu Valley.

Finally, it should be noted that these regression simulations assume a correct estimation of the temporal random error. Underestimating this error propagates in the fit of the regression slope. Figure 12 presents the possible effect of such an underestimation when the a priori random error of the reference measurements is set at $1x10^{15}$ molec $cm^{-2}$, using again the
AirMAP observations of Fig. 5 (right panel) as input data. As the dynamic range of the reference measurements increases with the applied error, the fitted slope decreases. For a true error of $4x10^{15}$, this leads for instance to an underestimation of the slope of about 5%. This effect is small but other sources of random error (e.g undersampling the satellite pixels) would add up in a real-world experiment. Wang et al. (2017) observed such a systematic decrease of the regression slope when averaging MAX-DOAS measurements within larger time windows around the satellite overpass.

**4.2 Lessons learned for the validation of space-borne $H_2CO$ VCDs**

Table 4 is similar to Table 3 but for $H_2CO$, which we only measured in significant amounts in and around Bucharest.

The background level of the $H_2CO$ VCD around the city is around $1x10^{16}$ molec $cm^{-2}$ and the anthropogenic increase in the city center is up to $7x10^{16}$ molec $cm^{-2}$ (Fig. 7). The background falls within the TROPOMI $H_2CO$ spread ($1.2x10^{16}$ molec $cm^{-2}$), and Fig. 7 indicates that the extent of the urban hotspot only corresponds to a few TROPOMI pixels, with a
maximum at 6 $\sigma$. This limits the relevance of individual mapping flights for the validation of $H_2CO$, yet systematic airborne measurements would improve the statistics. The information on the $H_2CO$ horizontal variability is nevertheless useful, as it justifies the installation of a second MAX-DOAS in the city center, in addition to background measurements outside the city. Indeed, long-term ground-based measurements at two sites would be useful to investigate seasonal variations of $H_2CO$, as already demonstrated in other sites (De Smedt et al., 2015). Averaging the $H_2CO$ over a season would reduce the random
errors of the satellite measurements and it could reveal the horizontal variability of $H_2CO$ from space. The $H_2CO$ hotspot around Bucharest seems to be visible in the TROPOMI data of summer 2018 (I. De Smedt, personnal communication).

Getting information on the profile of $H_2CO$ during an airborne campaign may also help to understand the differences between ground-based and space-borne observations. This could be done by adding to the BN-2 instrumental set-up an in-situ $H_2CO$ sensor such as the In Situ Airborne Formaldehyde instrument (ISAF, Cazorla et al. (2015)) or the COmpact Formalde-
hyde FluorescencE Experiment (COFFEE, St. Clair et al. (2017)).

**4.3 Lessons learned for the validation of space-borne $SO_2$ VCDs**

Table 5 is similar to Table 3 but for $SO_2$, which we only measured in significant amounts in the Jiu Valley. The higher bias of the airborne measurements for $SO_2$ compared to $NO_2$ is due to the albedo. The latter is lower in the UV where we retrieve $SO_2$, which leads, for the same albedo error, to a larger AMF uncertainty (e.g. Merlaud et al., 2018, Fig.10).
Averaging the $SO_2$ VCDs from the airborne mapping of Fig. 10 at the TROPOMI resolution leads to 30 near nadir TROPOMI pixels above a 2-$\sigma$ error of $5.4x10^{16}$ molec $cm^{-2}$. The maximum $SO_2$ tropospheric VCD seen by TROPOMI would be $2.4x10^{17}$ molec $cm^{-2}$ (7 $\sigma$). This tends to indicate that airborne mappings of $SO_2$ VCDs above large power plants could help to validate

the horizontal variability of the $SO_2$ VCDs measured from space, to a limited extent in the AROMAT conditions due to the small dynamic range (7 $\sigma$). As for $H_2CO$, systematic airborne measurements would improve the statistics.

However, it would be difficult to quantify the bias of the satellite $SO_2$ VCD with AROMAT-type of airborne measurements. Adding in quadrature the biases of the $SO_2$ VCDs for airborne measurements (40%, Table 5) and for TROPOMI (30%, Table 2) already leads to a combined uncertainty of 50%, without considering any temporal variation or regression error. This best-case scenario is already at the upper limit of the TROPOMI requirements for tropospheric $SO_2$ VCDs (Table 2).

Similar to $H_2CO$, the validation of the satellite-based $SO_2$ measurements should thus rely on ground-based measurements,
enabling to improve the signal-to-noise ratio of the satellite and reference measurements by averaging their time series. An additional difficulty for validating $SO_2$ VCDs emitted by a power plant arise from the spatial heterogeneity of the $SO_2$ field around the point source, which renders ground-based VCDs measurements complicated.

On the other hand, Fioletov et al. (2017) presented a method to derive the $SO_2$ emissions from OMI data and validated it against reported emissions. The $SO_2$ fluxes can be measured locally in several ways and we tested some of them during
AROMAT-2 (see Sect. 4.4.2 below). To validate satellite-derived $SO_2$ products in Europe, it thus seems possible to compare satellite and ground-based reference $SO_2$ fluxes. Theys et al. (2019) already validated TROPOMI-derived volcanic $SO_2$ fluxes against ground-based measurements. In this context, a $SO_2$ camera pointing to the plant stack would be a valuable tool since it could be permanently installed and automated. One advantage of such a camera compared to the other tested remote-sensing instruments, beside its low operating cost, is that it derives the extraction speed from the measurements, avoiding dependence
on low-resolution wind information. The next section presents the $SO_2$ fluxes derived with such a camera during the 2015 campaign.

Note that the $SO_2$ VCDs measured on 28 August 2015 around Turceni may be higher than in standard conditions due to a temporary shutdown of the desulfurization unit, which was reported by local workers. $SO_2$ VCDs in the area seem to have decreased (D. Constantin, personnal communication). The first reported TROPOMI $SO_2$ measurements above the area
pinpoint other power plants in Serbia, Bosnia–Herzegovina, and Bulgaria (Fioletov et al., 2020). For validation studies, it would be worth to install automatic $SO_2$ cameras around these plants, until they are equipped with FGD units.

## 4.4 Emissions of $NO_x$ and $SO_2$ from Bucharest and the Jiu Valley

This section presents estimates of the $NO_x$ and $SO_2$ fluxes from Bucharest and the power plants in the Jiu Valley, combining our different 2014 and 2015 measurements and comparing them with available reported emissions. Campaign-based estimates
of $NO_x$ emissions from large sources are relevant in a context of satellite validation since the high resolution of TROPOMI enables to derive such emissions on a daily basis (Lorente et al., 2019). Regarding $SO_2$, as discussed in the previous section, the low signal-to-noise ratio of the satellite measurements implies averaging for several months to derive a $SO_2$ flux (Fioletov et al., 2020), yet campaign measurements are useful to select an interesting site and test the ground-based apparatus and algorithms.

The comparisons with reported emissions should not be overintrepreted since we compare campaign-based flux measure-
ments performed during a few days in daytime with reported emissions which represent yearly averages. Nevertheless, they give interesting indications about the operations of the FGD units of the power plants and possible biases in emission inventories.

Our flux estimates are all based on optical remote sensing measurements. They involve integrating a transect of the plume along its spatial extent and multiplying the outcome by the plume speed, which may correspond to the stack exit velocity (camera pointing to the stack) or to the wind speed (Mobile-DOAS and imaging-DOAS). We refer the reader to previous studies for the practical implementations. Ibrahim et al. (2010) presented the method we used for Bucharest, where we encircled the city with the Mobile-DOAS. Meier et al. (2017) presented the AirMAP-derived flux estimations, while Johansson et al. (2014) derived industrial emissions from a car-based Mobile-DOAS instrument as we did for the Turceni power plant. Constantin et al. (2017) presented the fluxes based on the ULM-DOAS measurements. Regarding the $SO_2$ cameras, they are now commonly used to monitor $SO_2$ emissions from volcanoes (see McGonigle et al. (2017) and references therein), but their capacity to measure $SO_2$ fluxes from power plants have been demonstrated as well (e.g., Smekens et al., 2014).

### 4.4.1 $NO_x$ flux from Bucharest

We estimated $NO_x$ fluxes from the Bucharest urban area using the $NO_2$ VCDs measured with the UGAL Mobile-DOAS systems along the external ring and the wind data on 8 September 2014 and 31 August 2015. We derived the wind direction from the maxima of the $NO_2$ VCDs in the DOAS observations. For the wind speed, we took 1.1 m s$^{-1}$ on 8 September 2014, the value Meier (2018) used for the AirMAP-derived flux, which originates from meteorological measurements at Baneasa airport. On 31 August 2015, we used the ERA5 wind data (C3S, 2017) at the time when the Mobile-DOAS crossed the $NO_2$ plume (15:00 UTC). The ERA5 database indicates a constant windspeed between 1000 and 900 hPA of 1.2 m s$^{-1}$. Finally and similarly to Meier (2018), we took a ratio of 1.32 for the $NO_x$ to $NO_2$ ratio and estimated the chemical loss of $NO_x$ with a lifetime of 3.8h and an effective source location in the center of Bucharest.

Table 7 presents the AirMAP and Mobile-DOAS derived $NO_x$ fluxes from Bucharest, ranging between 12.5 and 17.5 mol.s$^{-1}$. On 8 September 2014, the Mobile and airborne observations were coincident. Their estimated $NO_x$ fluxes agree within 20%. This gives confidence in the flux estimation yet one should keep in mind that the same wind data was used for both estimations. Meier (2018) estimated the uncertainties on the AirMAP-derived $NO_x$ flux to be around 63%, while the uncertainty of Mobile-DOAS derived $NO_x$ flux typically range between 30% and 50% (Shaiganfar et al., 2017).

We compared our measured $NO_x$ fluxes with the European Monitoring and Evaluation Programme inventory (EMEP, https://www.ceip.at/). In practice, we summed the EMEP gridded yearly $NO_x$ emissions between 44.2°and 44.6°N and between 25.9 °E and 26.3 °E and we assumed the emissions are constant during one year. This led to $NO_x$ emissions of 6.14 and 6.33 mol s$^{-1}$ for 2014 and 2015. Studying the reported emissions from several European cities including Bucharest, Trombetti et al. (2018) mentions that the EMEP emissions are well below other inventories for all the pollutants. We thus also compared our flux with the Emissions Database for Global Atmospheric Research (EDGAR v4.3.2, Crippa et al. (2018)), which is only available until 2012. The same method led to a $NO_x$ flux of 18.4 mol s$^{-1}$, to compared with the 2012 EMEP $NO_x$ emissions of 7.1 mol s$^{-1}$. Based on summer measurements, the AROMAT-derived $NO_x$ emissions do not include residential heating. The latter ranges between 10 and 40% of the total $NO_x$ according to Trombetti et al. (2018). This tends to confirm that the EMEP inventory underestimates the $NO_x$ emissions for Bucharest.

### 4.4.2 NO$_x$ and SO$_2$ fluxes from the power plants in the Jiu Valley

Figure 13 presents a scatter plot of the slant columns of NO$_2$ and SO$_2$ for the ultralight flight of 26 August 2015, which detected the four exhaust plumes of the Valley between 08:31 and 11:04 UTC. Two regimes are visible in the SO$_2$ to NO$_2$ ratio. When considering the longitude, the low SO$_2$ to NO$_2$ ratio (1.33) appears to correspond to the Rovinari exhaust plume, while the other power plants exhibit a higher ratio (13.55). The low ratio observed at Rovinari corresponds to the FGD units operating at this power plant.

We estimated the NO$_x$ and SO$_2$ flux from the power plants using several instruments: a Mobile-DOAS, the ULM-DOAS, and the SO$_2$ camera. For the DOAS instruments, we inferred the wind direction from the plume position and we retrieved the wind speed from the ERA5 database. Considering the observed vertical extent of the plume downwind of Turceni (Fig. 9), we took the wind speed at 950 hPa (ca. 500 m a.s.l.).

Figure 14 presents the ULM-DOAS-estimated fluxes of NO$_x$ and SO$_2$ from the power plants in Turceni, Rovinari, and Craiova for the flight on 26 August 2015. The figure also shows the reported emissions from the European Environment Agency (EEA) large combustion plants database (EEA, 2018), assuming constant emissions throughout the year. Turceni appears to be the largest SO$_2$ source (78 mol s$^{-1}$), while Rovinari is the largest NO$_x$ source (8 mol s$^{-1}$).

It is difficult to interpret the discrepancies between those measured fluxes and the yearly reported emissions since we observed large variations in the instantaneous emissions with the SO$_2$ camera (see below and Fig. 15). However, the ratio of the two fluxes appears interesting since we can assume its relative stability. This ratio for a given power plant depends on whether or not a desulfurization unit is operational at the plant. On Fig. 14, Turceni appears to have both the largest measured ratio and the largest discrepancy between the measured and reported ratios. This is consistent with a temporary shutdown of the desulfurization unit of the Turceni power plant, as was reported by the plant workers during the campaign. The ULM-DOAS measurements on 25 August 2015 (shown in Fig. S11 the Supplement), which also sampled the Isalnita plume, are consistent with those of 26 August 2015. These measurements enable to estimate total NO$_x$ and SO$_2$ fluxes to be about 22 and 147 mol s$^{-1}$, respectively.

Table 8 focuses on the Turceni power plant and lists all estimates of the NO$_x$ and SO$_2$ emissions from this source. Meier (2018) estimated the NO$_x$ flux from the Turceni power plant using the AirMAP measurements of 2014 and 2015. This leads to similar values for the two flights on 11 September 2014 and 28 August 2015, of about 8 mol.s$^{-1}$. On this second day, the UGAL Mobile-DOAS crossed the plume along the road in front of the power plant. These ground-based measurements lead to a NO$_2$ flux of 2.2 mol.s$^{-1}$, much lower than the aforementioned AirMAP-derived value. However, Meier (2018) calculated the latter based on AirMAP measurements at 3.5 km from the source. At shorter distances, the AirMAP estimated NO$_2$ flux is smaller and close to the Mobile-DOAS observations. This is probably related to the fact that the NO/NO$_2$ ratio has not yet reached its steady state value above the road where we performed the Mobile-DOAS observations, which is only around 1 km from the stack. The agreement is better for SO$_2$ (25 and 32 mol.s$^{-1}$). On 25 August 2015, we had a coincidence of ULM-DOAS and Mobile-DOAS observations and we observed a similar range of values. This gives us confidence in our estimate of the NO$_x$ flux from the aircraft but confirms that the nearby road is too close to the plant to estimate a meaningful NO$_x$ flux

from Mobile-DOAS $NO_2$ observations. Note that the conversion of NO into $NO_2$ is also visible right above the Turceni stack in the $NO_2$ imager data of 24 August 2015, as appears in Fig.6 of Dekemper et al. (2016).

Figure 15 presents a time series of the $SO_2$ emissions from the Turceni power plant between 9:00 and 10:50 UTC on 28 August 2015. We derived $SO_2$ fluxes at different altitudes above the stack using a UV $SO_2$ camera which is an updated version of the Envicam2 system, used during the $SO_2$ camera intercomparison described by Kern et al. (2015). We converted the measured optical densities to $SO_2$ column densities using simultaneous measurements with an integrated USB spectrometer (Lübcke et al., 2013). We estimated the stack exit velocity from the $SO_2$ images, recorded with a time resolution of about 15 seconds, by tracking the spatial features of the plume. Dekemper et al. (2016) used a similar approach to derive the $NO_2$ flux from $NO_2$ camera imagery.

The $SO_2$ fluxes retrieved for transverses at 400 to 700 m vertical distances above the stack agree on average with each other within 20%. Emissions estimated 100 m above the stack are underestimated due to saturation ($SO_2$ column densities above 2 x $10^{18}$ molec.cm$^{-2}$) and high aerosol concentration close to the exhaust.

The $SO_2$ emissions show large fluctuations. During the time of our observations they increased from 1 kg.s$^{-1}$ (15.6 mol.s$^{-1}$) to around 4 $\pm$ 1 kg$^{-1}$ (62.4 mol.s$^{-1}$). The images (Fig. S10 in the Supplement) also show a second and weaker source that emits $SO_2$. This is probably the desulfurization unit, which was reported to be turned on again on this day, after the temporary shutdown. Indeed, as appears in Table 8, the $SO_2/NO_2$ ratio measured from AirMAP is lower than the ones measured from the ULM-DOAS during the previous days, and the same holds true for the Mobile-DOAS measurements.

## 5   Conclusions

The two AROMAT campaigns took place in Romania in September 2014 and August 2015. They combined airborne and ground-based atmospheric measurements and focused on air quality-related species ($NO_2$, $SO_2$, $H_2CO$, and aerosols). The AROMAT activity targeted the urban area of Bucharest and the power plants of the Jiu Valley. The main aims were to test new instruments, measuring the concentrations and emissions of key pollutants in the two areas, and investigating the concept of such campaigns for the validation of air quality satellite-derived products.

We have shown that the airborne mapping of tropospheric $NO_2$ VCDs above Bucharest is potentially valuable for the validation of current and future nadir-looking satellite instruments. In the AROMAT conditions, airborne measurements were consistent with ground-based observations within 7% and covered a significant part of the dynamic range of the $NO_2$ tropospheric VCDs at an appropriate signal to noise ratio. Our simulations, based on campaign measurements and TROPOMI characteristics, indicate that we can constrain the accuracy of the satellite $NO_2$ VCDs within 28 or 37%, depending on whether information on the aerosol and $NO_2$ profile is available or not. This points to the importance of acquiring profile information to approach the TROPOMI optimal target accuracy for tropospheric $NO_2$ VCDs (25%).

A unique advantage of airborne mapping is its ability to validate the imaging capabilities of nadir-looking satellites. This feature becomes more important as the satellite horizontal resolutions reaches the suburban scale. Judd et al. (2019) pointed out the difficulty for static ground-based measurements to represent the $NO_2$ VCDs measured from space in polluted areas, due to

the horizontal representativeness error. This error cancels out by mapping the full extent of satellite pixels. The caveat is the temporal error, which can be larger than with static ground-based measurements. For a single morning flight above Bucharest, we have estimated the random part of this temporal error to be about 4 x $10^{15}$ molec cm$^{-2}$. In the AROMAT conditions, underestimating this error would lead to a low bias in the regression slope between satellite and airborne measurements. This temporal error varies with local conditions for a given experiment but the satellite air quality community should further investigate this effect. This indicates the usefulness of simultaneous ground-based measurements, which may also be useful to estimate the reference NO$_2$ VCDs in the airborne observations. These conclusions for NO$_2$ above Bucharest apply to other large polluted urban areas.

In addition to NO$_2$, we also detected the signature of H$_2$CO emissions in and around Bucharest, with an anthropogenic hotspot in the city center. Due to the lower signal to noise ratio of the spaceborne H$_2$CO observations, it is difficult to use such daily measurements for satellite validation. We thus propose considering long-term ground-based MAX-DOAS measurements in the city for the validation of H$_2$CO.

In the Jiu Valley, NO$_2$ is clearly visible from both satellite and aircraft, and the VCDs are comparable in magnitude with the signal detected above Bucharest. However, it appears more complicated to quantitatively compare the NO$_2$ VCDs datasets in the thick exhaust plumes of the power plants. These plants also emit SO$_2$ but, as for H$_2$CO, the low signal to noise ratio of satellite measurements reduces the validation relevance of individual airborne measurements.

In relation to the ideal validation study mentioned in the introduction, the relevance of international airborne campaigns is generally limited by its timespan of typically a couple of weeks, imposed by logistical and cost considerations. To overcome this limitation, we propose to consider routine airborne mapping of NO$_2$ VCDs by local aircraft operators and close to a well-equipped ground-based observatory. Such a set-up would reduce the fixed costs of the observations, which could then be allocated to flight hours in different seasons. Such an approach would combine the advantages of long-term ground-based and airborne measurements. In the longer term, high altitude pseudo-satellites (HAPS) could provide the necessary routine measurements above selected supersites, as needed to validate the observations from future sensors in geostationary orbit.

*Competing interests.* The authors declare that they have no conflict of interest.

*Author contributions.* AM, LB, D-EC, MDH, ACM, LG, DN, and MVR planned and organised the campaign. All coauthors contributed to the campaign either as participants or during campaign preparation and/or follow up data analysis, including the writing of this manuscript, which was coordinated by AM and MVR with feedback and contributions from all the coauthors.

*Acknowledgements.* The AROMAT activity was supported by ESA (contract $4000113511/15/NL/FF/gp$) and by the Belgian Science Policy (contract $BR/121/PI/UAV\,Reunion$). Regarding the AirMAP instrument, financial support through the University of Bremen

Institutional Strategy Measure M8 in the framework of the DFG Excellence Initiative is gratefully acknowledged. Part of the work performed for this study was funded by the Romanian Ministry of Research and Innovation through Program I - Development of the national research-development system, Subprogram 1.2 - Institutional Performance - Projects of Excellence Financing in RDI, Contract No.19PFE / 17.10.2018 and by Romanian National Core Program Contract No.18N/2019. Katharina Riffel supported the MPIC mobile DOAS measurements. We thank Klaus Pfeilsticker, Isabelle De Smedt, Nicolas Theys, Ermioni Dimitropoulou, Lori Neary, and the two anonymous referees for useful discussions. We also thank the people of Turceni and the Air Traffic Control of Romania for their support and cooperation.

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

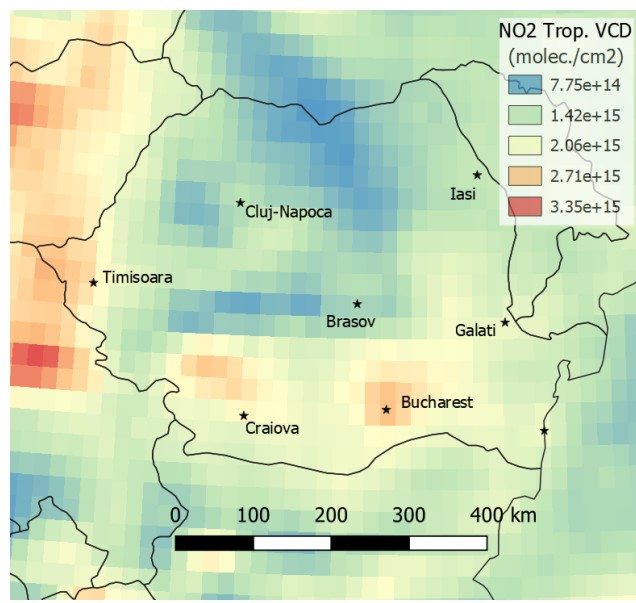

**Figure 1.** The tropospheric NO$_2$ VCD field seen from space with the OMI/AURA instrument above Romania (OMNO2d product, averaged for 2012-2016 with Giovanni, NASA GES DISC). The black stars pinpoint the largest cities of Romania.

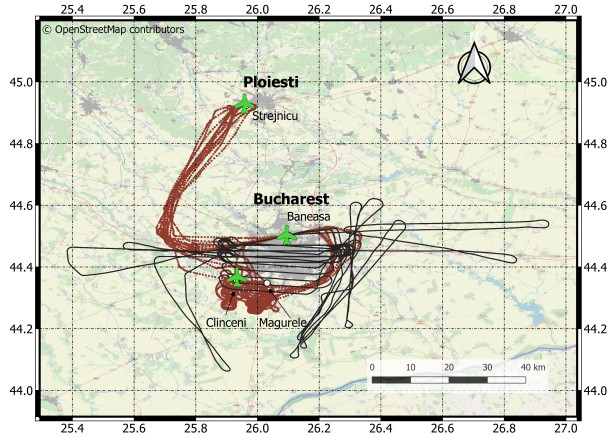

**Figure 2.** The Bucharest area with important locations for the AROMAT campaigns: the INOE atmospheric observatory in Magurele, the Baneasa airport, and the Clinceni airfield. Buit-up areas appear in grey. The red and black lines, respectively, show the BN-2 and Cessna flight tracks during AROMAT-2.

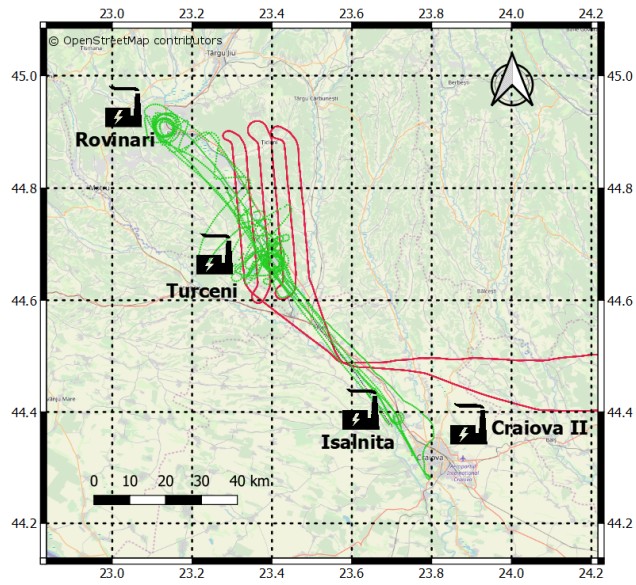

**Figure 3.** The Jiu Valley and its four power plants between Targu Jiu and Craiova. The scientific crew was based in Turceni during the AROMAT campaigns. The green and red lines, respectively, show the ultralight and Cessna flight tracks during AROMAT-2.

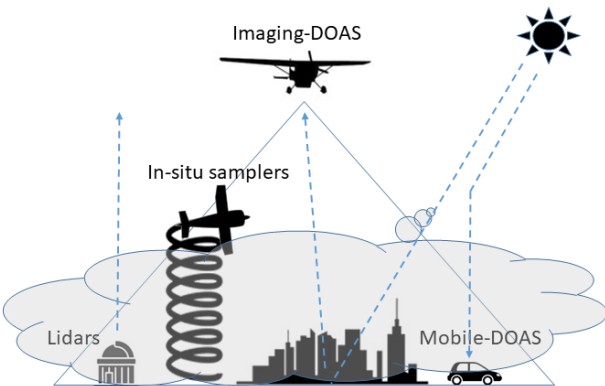

**Figure 4.** Geometry of the main measurements performed during the AROMAT campaigns. The Imaging-DOAS instruments map the $NO_2$ and $SO_2$ VCDs at 3 km altitude above the target area while the in-situ samplers measure profiles of trace gases and aerosols. Ancillary ground measurements include Mobile-DOAS to quantify trace gases VCDs and lidars to measure the aerosol optical properties.

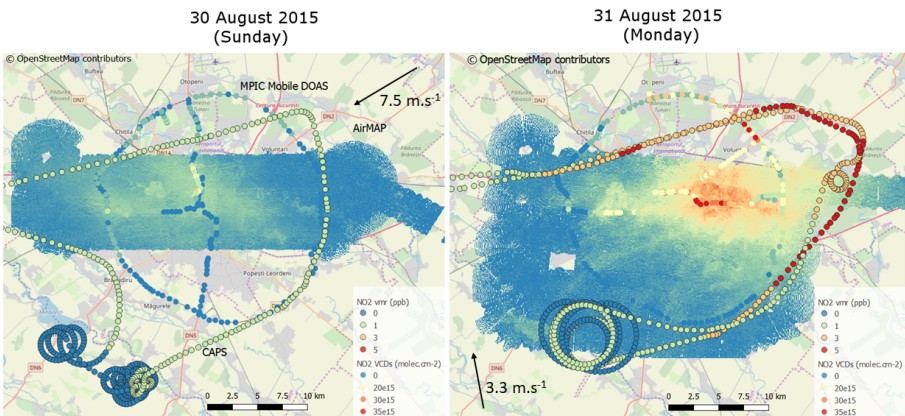

**Figure 5.** Measurements of $NO_2$ VCDs and volume mixing ratios in Bucharest on 30 (Sunday) and 31 (Monday) August 2015 with AirMAP (continuous map), the CAPS (black-rimmed circles), and the MPIC Mobile-DOAS (plain colour circles).

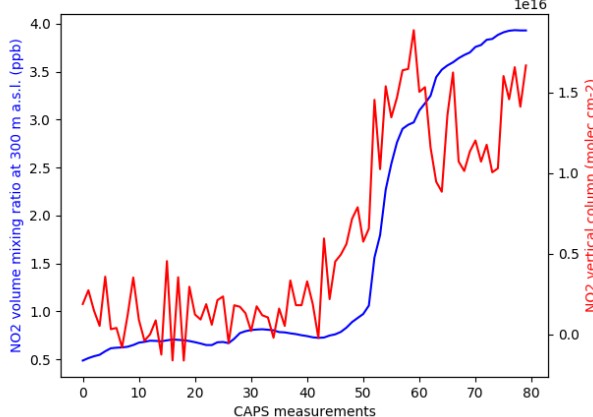

**Figure 6.** Volume mixing ratio and VCDs of $NO_2$ in and out of the pollution plume of Bucharest, as measured with the CAPS (on the BN-2, 12:30-12:55 UTC) and AirMAP (on the Cessna, 12:00-13:30 UTC) during the afternoon flights on 31 August 2015. Note that the plot shows the VCDs extracted at the position of the CAPS measurements.

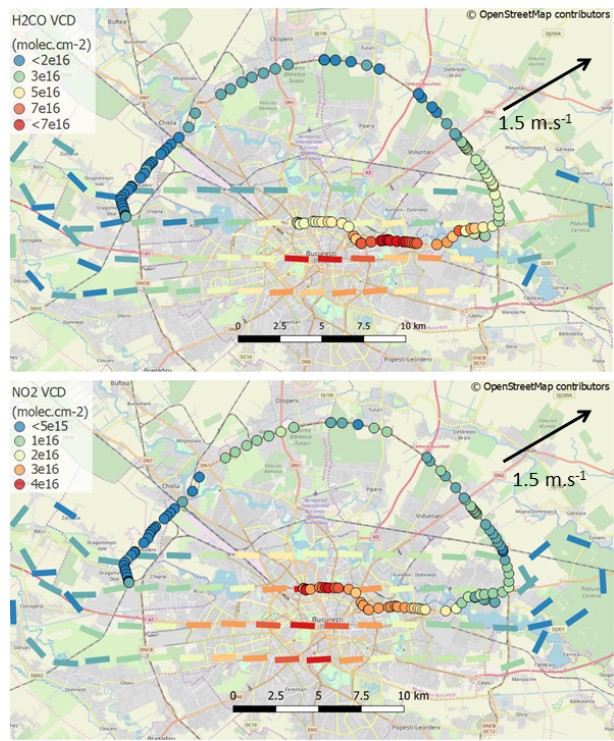

**Figure 7.** Horizontal distribution of tropospheric $H_2CO$ and $NO_2$ VCDs measured on 31 August 2015 with the IUP-Bremen nadir-only compact spectrometer from the Cessna (flight tracks, 07:46-08:23 UTC) and with the MPIC Mobile-DOAS (coloured circles, 08:13-10:00 UTC).

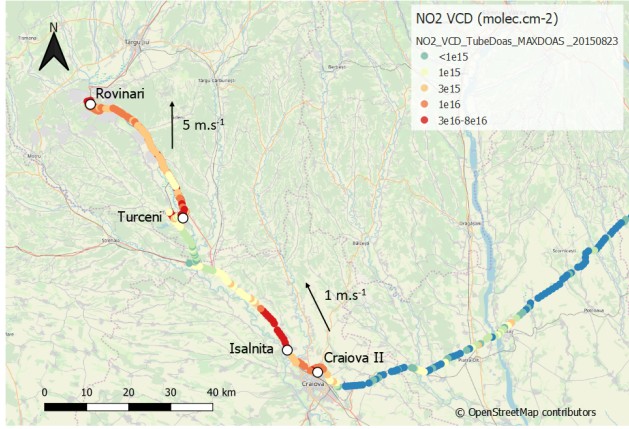

**Figure 8.** Tropopsheric vertical column densities of $NO_2$ measured with the MPIC Mobile-DOAS instruments in the Jiu Valley on 23 August 2015 between 08:07 and 14:16 UTC.

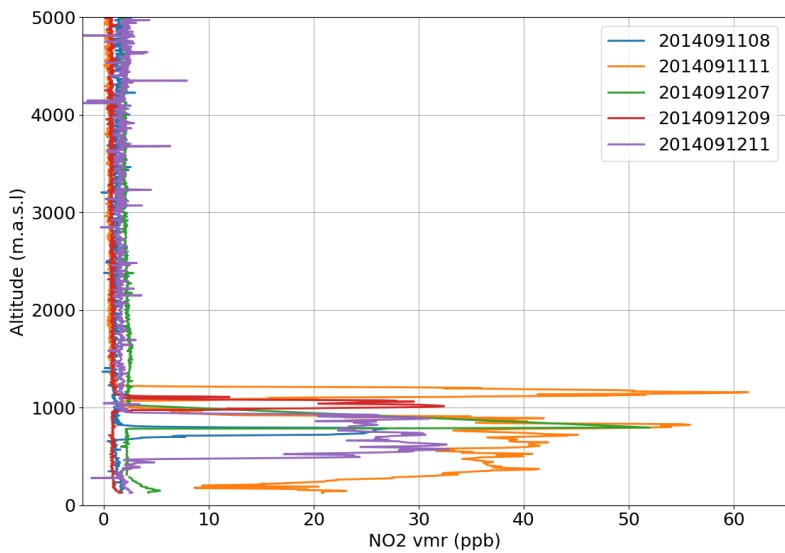

**Figure 9.** Examples of $NO_2$ sondes data from Turceni during AROMAT-1 (11 and 12 September 2014). The legend indicates the date, the last two digits being the hour of launch (UTC).

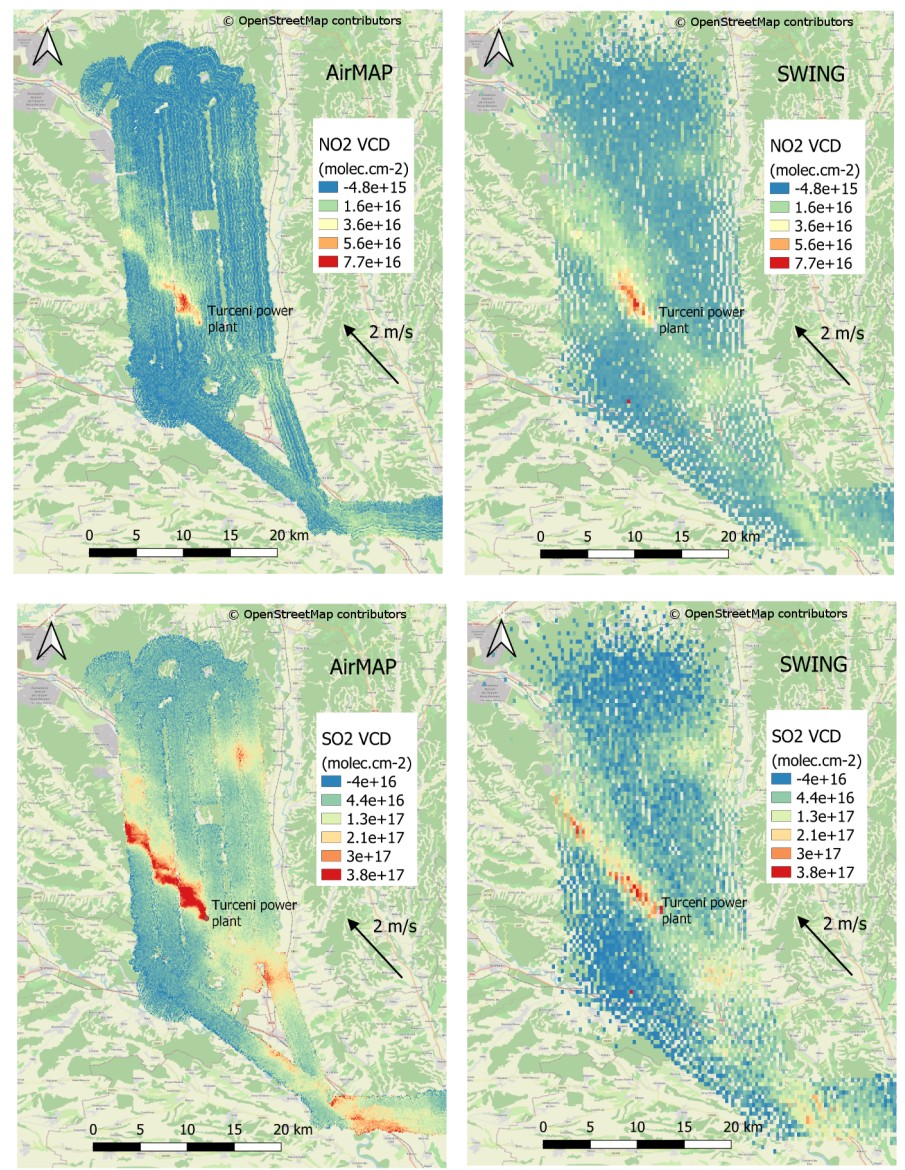

**Figure 10.** AirMAP (left panels) and SWING (right panels) $NO_2$ (upper panels) and $SO_2$ (lower panels) VCDs above Turceni (28 August 2015).

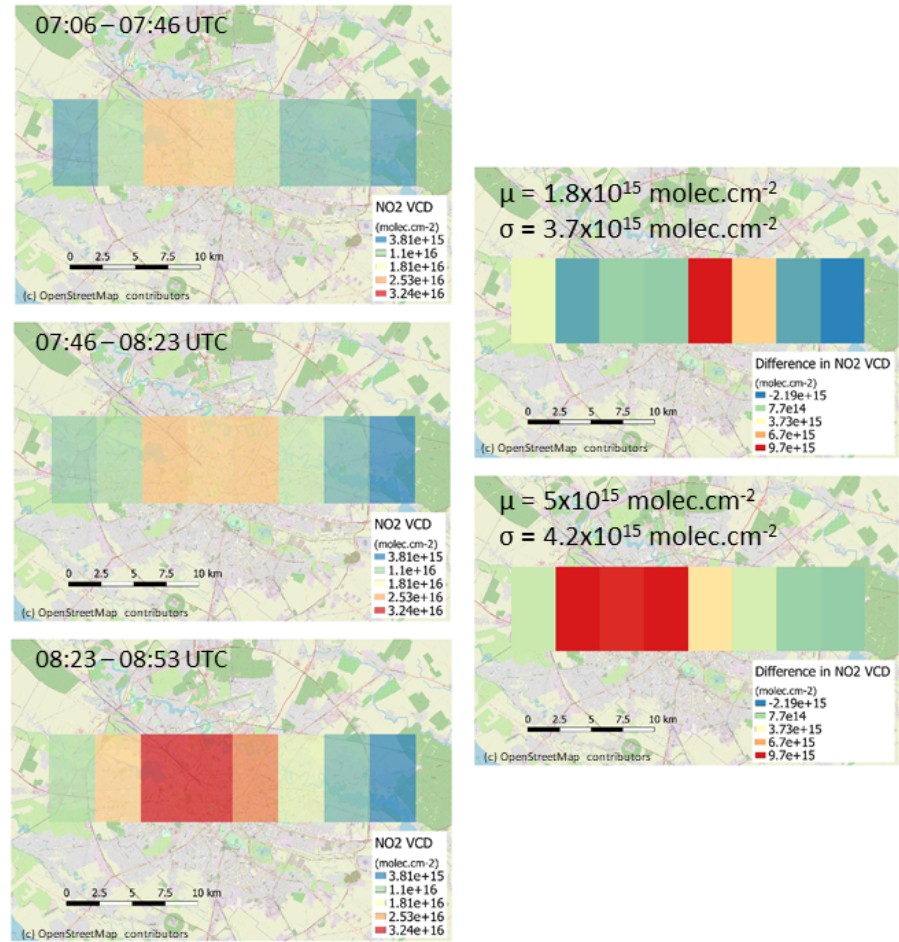

**Figure 11.** AirMAP measurements of $NO_2$ VCDs degraded at the TROPOMI resolution during three overpasses of the morning flight of 31 August 2015 (left panels), together with the differences of these degraded $NO_2$ VCDs for consecutive overpasses (right panels). The right panels also indicate the means ($\mu$) and standard deviations ($\sigma$) of the two differences.

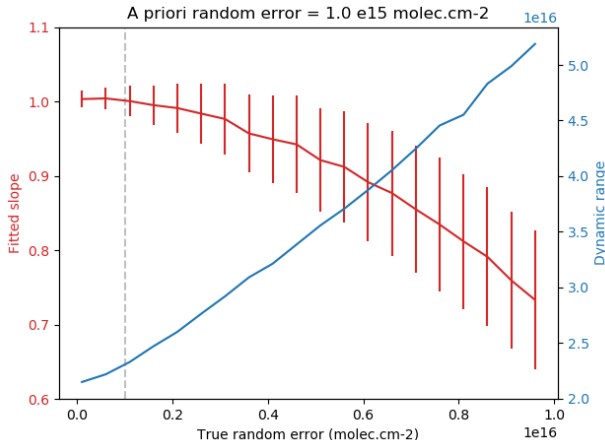

**Figure 12.** Effect of an underestimation of the random error in a regression analysis simulating TROPOMI validation using airborne mapping as reference measurements of $NO_2$ VCDs. The dynamic range (blue line) of the reference measurements increases with the applied random error. For the considered a priori random error (dashed vertical line, $1 \times 10^{15}$ molec.cm$^{-2}$), this leads to an underestimation of the regression slope (red line). These simulations use the AirMAP data of 31 August 2015 (afternoon flight).

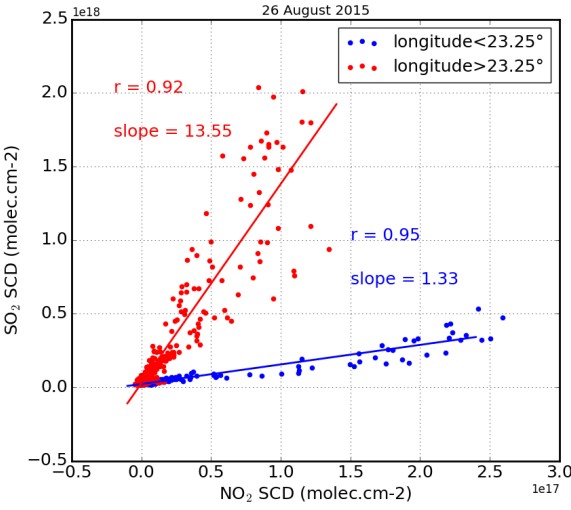

**Figure 13.** $SO_2$ and $NO_2$ SCDs SCDs measured from the ULM-DOAS above the Jiu Valley on 26 August 2015 between 08:31 and 11:04 UTC. Blue dots indicate the measurements above Rovinari, whereas the red ones are for all the other plants.

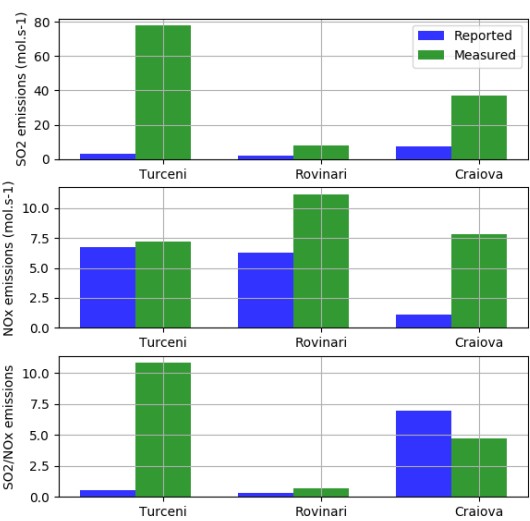

**Figure 14.** SO$_2$ and NO$_x$ fluxes from three power plants of the Jiu Valley as (1) measured with the ULM-DOAS on 26 August 2015 (green bars) and (2) estimated from the reported emissions of 2015 assuming constant emissions throughout the year (blue bars). Uncertainties on the ULM-DOAS-derived fluxes are around 60%.

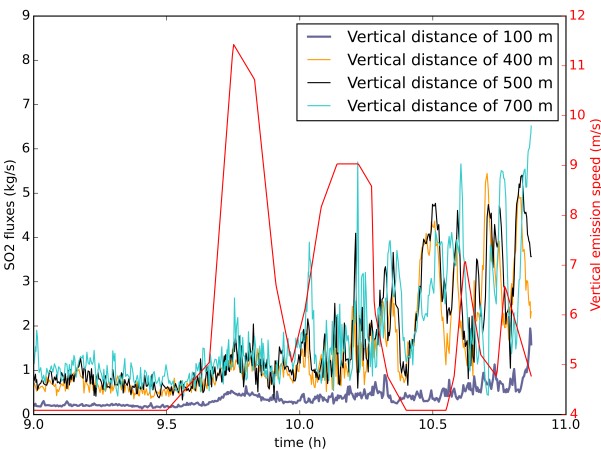

**Figure 15.** SO$_2$ fluxes from the Turceni power plant on 28 August 2015. They were estimated with the Envicam2 SO$_2$ camera for 4 transverses at vertical altitudes above the stack of 100, 400, 500 and 700 m. The red line shows the estimated plume speed (m.s$^{-1}$).

**Table 1.** Past and near-future space missions focused on air quality: coverage, pixel size, and temporal sampling.

| Launch year | Instrument | Pixel size at nadir (km$^2$) | Coverage | Revisit time |
|---|---|---|---|---|
| 1995 | GOME | 320 x 40 | Global | 3 days |
| 2002 | SCIAMACHY | 60 x 30 | Global | 6 days |
| 2004 | OMI | 13 x 24 | Global | 1 day |
| 2006 | GOME-2 | 80 x 40 | Global | 1 day |
| 2011 | OMPS | 50 x 50 | Global | 1 day |
| 2017 | TROPOMI | 3.5 x 5.5 | Global | 1 day |
| 2023 (planned) | Sentinel-5 | 7 x 7 | Global | 1 day |
| 2020 | GEMS | 7 x 8 | East Asia | 1 hour |
| 2022 (planned) | TEMPO | 2 x 4.5 | North America | 1 hour |
| 2023 (planned) | Sentinel-4 | 8 x 8 | Europe | 1 hour |

**Table 2.** Data quality targets for the S5P TROPOMI data products relevant in the AROMAT context (extracted from ESA (2014)).

| Product | Accuracy | Precision |
|---|---|---|
| Tropospheric $NO_2$ | 25-50% | $7 \times 10^{14}$ molec cm$^{-2}$ |
| Tropospheric $SO_2$ | 30-50% | $2.7\text{-}8.1 \times 10^{16}$ molec cm$^{-2}$ |
| Total $H_2CO$ | 40-80% | $0.4\text{-}1.2 \times 10^{16}$ molec cm$^{-2}$ |

**Table 3.** Summary of the AROMAT measurements of $NO_2$.

| Instrument | Type | Ground Sampling Distance (m) | Observed range (molec cm$^{-2}$ / ppb) | Detection limit (molec cm$^{-2}$ / ppb) | Bias (%) | Reference |
|---|---|---|---|---|---|---|
| AirMAP | Imager | 100 | $0\text{-}8 \times 10^{16}$ | $1.5 \times 10^{15}$ | 25% | Meier et al. (2017) |
| SWING | Imager | 300 | $0\text{-}8 \times 10^{16}$ | $1.2 \times 10^{15}$ | 25% | Merlaud et al. (2018) |
| ULM-DOAS | Nadir | 400 | $0\text{-}1.7 \times 10^{17}$ | $5 \times 10^{14}$ | 25% | Constantin et al. (2017) |
| IUP-Bremen nadir | Nadir | 1800 | $0\text{-}3.5 \times 10^{16}$ | $2 \times 10^{15}$ | 25% | Bösch et al. (2016) |
| Tube MAX-DOAS | Car-based | 500 | $0\text{-}1.3 \times 10^{17}$ | $1.3 \times 10^{14}$ | 20% | Donner et al. (2015) |
| Mini Max-DOAS | Car-based | 500 | $0\text{-}1.3 \times 10^{17}$ | $6 \times 10^{14}$ | 20% | Wagner et al. (2010) |
| UGAL Mobile | Car-based | 500 | $0\text{-}2.5 \times 10^{17}$ | $4 \times 10^{14}$ | 25% | Constantin et al. (2013) |
| BIRA Mobile | Car-based | 500 | $0\text{-}1.3 \times 10^{17}$ | $8 \times 10^{14}$ | 20% | Merlaud (2013) |
| KNMI sonde | In-situ | n.a. | 0-60 | 1 | 40% | Sluis et al. (2010) |
| CAPS | In-situ | n.a. | 0-20 | 0.1 | 40% | Kebabian et al. (2005) |

**Table 4.** Summary of the AROMAT measurements of $H_2CO$.

| Instrument | Type | Ground Sampling Distance (m) | Observed range (molec cm$^{-2}$) | Detection limit (molec cm$^{-2}$) | Bias (%) | Reference |
|---|---|---|---|---|---|---|
| IUP-Bremen nadir | Nadir | 1800 | $1\text{-}7 \times 10^{16}$ | $6 \times 10^{15}$ | 25% | Bösch et al. (2016) |
| Tube MAX-DOAS | Car-based | 500 | $1\text{-}7.5 \times 10^{16}$ | $8 \times 10^{14}$ | 20% | Donner et al. (2015) |

**Table 5.** Summary of the AROMAT measurements of $SO_2$.

| Instrument | Type | Ground Sampling Distance (m) | Observed range (molec cm$^{-2}$) | Detection limit (molec cm$^{-2}$) | Bias (%) | Reference |
|---|---|---|---|---|---|---|
| AirMAP | Imager | 100 | $0\text{-}6 \times 10^{17}$ | $1.7 \times 10^{16}$ | 40% | Meier et al. (2017) |
| SWING | Imager | 300 | $0\text{-}4 \times 10^{17}$ | $2 \times 10^{16}$ | 40% | Merlaud et al. (2018) |
| ULM-DOAS | Nadir | 400 | $0\text{-}2.5 \times 10^{18}$ | $3 \times 10^{15}$ | 40% | Constantin et al. (2017) |
| Tube MAX-DOAS | Car-based | 500 | $0\text{-}1 \times 10^{18}$ | $5 \times 10^{15}$ | 20% | Donner et al. (2015) |
| Mini Max-DOAS | Car-based | 500 | $0\text{-}2.2 \times 10^{18}$ | $1 \times 10^{16}$ | 20% | Wagner et al. (2010) |
| UGAL Mobile | Car-based | 500 | $0\text{-}4 \times 10^{18}$ | $4 \times 10^{15}$ | 25% | Constantin et al. (2013) |

**Table 6.** Total simulated error budget for the validation of spaceborne $NO_2$ VCDs validation using airborne mapping at different resolution, with or without profile informations.

| | | Precision (molec cm$^{-2}$) | | | Accuracy | | |
|---|---|---|---|---|---|---|---|
| | Place | Shot noise | Time error | Tot. | Ref. | Fit | Tot. |
| AirMAP | B | $3 \times 10^{13}$ | $4 \times 10^{15}$ | $4.1 \times 10^{15}$ | 26% | 6% | 37% |
| SWING | B | $7 \times 10^{13}$ | $4 \times 10^{15}$ | $4.1 \times 10^{15}$ | 26% | 6% | 37% |
| AirMAP + profile | B | $3 \times 10^{13}$ | $4 \times 10^{15}$ | $4.1 \times 10^{15}$ | 10% | 6% | 28% |
| AirMAP | T | $3 \times 10^{13}$ | $4 \times 10^{15}$ | $4.1 \times 10^{15}$ | 26% | 10% | 37% |
| AirMAP + profile | T | $3 \times 10^{13}$ | $4 \times 10^{15}$ | $4.1 \times 10^{15}$ | 10% | 10% | 29% |

**Table 7.** $NO_x$ emissions from Bucharest estimated from the AROMAT measurements. Note that we respectively use UGAL and MPIC Mobile-DOAS measurements for the estimates on 8 September 2014 and 31 August 2015.

| | AirMAP | Mobile-DOAS |
|---|---|---|
| 8 September 2014 | 14.6 mol.s$^{-1}$ | 12.5 mol.s$^{-1}$ |
| 9 September 2014 | 13.1 mol.s$^{-1}$ | n.a. |
| 31 August 2015 | n.a. | 17.5 mol.s$^{-1}$ |

**Table 8.** $NO_x$ and $SO_2$ emissions from the Turceni power plant estimated from the AROMAT measurements.

| | Instrument | Distance | $SO_2$ flux | $NO_x$ flux | $\frac{SO_2}{NO_2}$ |
|---|---|---|---|---|---|
| 11 September 2014 - 09:00 UTC | AirMAP | 7 km | n.a. | 8 mol.s$^{-1}$ | n.a. |
| 25 August 2015 - 07:45 UTC | Mobile-DOAS | 1 km | 105 mol.s$^{-1}$ | 4 mol.s$^{-1}$ | 15.4 |
| 25 August 2015 - 08:30 UTC | Mobile-DOAS | 1 km | 52 mol.s$^{-1}$ | 2 mol.s$^{-1}$ | 26 |
| 25 August 2015 - 08:30 UTC | ULM-DOAS | 10 km | 85 mol.s$^{-1}$ | 10 mol.s$^{-1}$ | 8.5 |
| 26 August 2015 - 10:00 UTC | ULM-DOAS | 5 km | 78 mol.s$^{-1}$ | 6 mol.s$^{-1}$ | 13 |
| 27 August 2015 - 07:45 UTC | ULM-DOAS | 8.5 km | 145 mol.s$^{-1}$ | 17 mol.s$^{-1}$ | 8.5 |
| 27 August 2015 - 07:55 UTC | Mobile-DOAS | 1 km | 77 mol.s$^{-1}$ | 5 mol.s$^{-1}$ | 16 |
| 28 August 2015 - 07:00 UTC | Mobile-DOAS | 1 km | 24.8 mol.s$^{-1}$ | 1.7 mol.s$^{-1}$ | 14.7 |
| 28 August 2015 - 10:00 UTC | AirMAP | 7 km | 25 mol.s$^{-1}$ | 8 mol.s$^{-1}$ | 3.1 |
| 28 August 2015 - 10:15 UTC | Mobile-DOAS | 1 km | 32 mol.s$^{-1}$ | 4 mol.s$^{-1}$ | 8 |
| 28 August 2015 - 09:00-11:00 UTC | $SO_2$ camera | Above stack | 15.6-62.4 mol.s$^{-1}$ | n.a | n.a. |