# Peer review of "Satellite validation strategy assessments based on the AROMAT campaigns"

_Atmospheric Measurement Techniques, 2019_

## Referee Comment (RC1) · Anonymous Referee #1 · 6 Mar 2020

Comments for the AMTD Manuscript, 'The Airborne ROmanian Measurements of Aerosols and Trace gases (AROMAT) campaigns'

This paper discusses the air quality measurement campaign, AROMAT, and puts it into the context of if/how these measurements can assist in future validation efforts for satellites, such as Sentinel 5P TROPOMI. The significance of this work is within the scope of AMT and is key as the air quality community works toward validating satellites that measure urban air quality (e.g., TROPOMI) and there are some novel 'take-home' messages from this work that are worthy for publishing. From what is shared in the paper, the quality of the work appears valid however there is clarity needed in some areas. This paper also needs restructuring to improve the clarity of the take-home conclusions. For example: The paper lacks details about the campaign and information about the measurements are scattered throughout the paper.

Specific comments/questions:
- The title of this paper does not clearly reflect the contents of the paper. The current title would attract readers as an overview paper for the measurements during AROMAT, but this is not the purpose of this paper nor is there a detailed overview of the entire campaigns. If agreed by the authors, I suggest changing to title to something that reflects that AROMAT could be a concept model for validation campaigns of satellite retrievals.
- Are conclusions made about validation only valid over Romania? Or can these lessons be extended beyond Romania? Please be clear in the paper which conclusions can be extended beyond the AROMAT region.
- It seems that the model for the conclusions is based on TROPOMI requirements. Please comment on if/how this extends to the requirements of the other planned missions or make the specific message in the paper that the conclusions that are made are specific to TROPOMI.
- A weakness in the general analysis is the lack of discussion on temporal variation and the time of the airborne and ground based measurements and how this relates to the time of the satellite overpass, emissions inventories, etc. The authors should keep this in mind to address through revisions.
- Section 2 should start by painting a picture of AROMAT 1 and 2 deployments and measurements that used in this analysis. While much of this information is in the supplement and scattered throughout the paper, the general reader enters Section 3 without the proper background to assess what is being discussed. Currently, it does not effectively communicate the needed details about the AROMAT campaign before moving into the results sections. To fix this, the authors could reorganize the section by moving 2.3 to before 2.1 and 2.2. Then there needs to be discussion (and maybe a Table) that summarizes each campaign. This table and/or discussion must include time periods of each deployment, location of each deployment, payloads for the aircraft and relevant details about ground measurements (in line with Tables 4-6) and could extend into partners and other details from 2.3 as seen fit.
  - Table 3 does not add substantial information to this paper and that space would be more effectively be used to summarize the campaigns themselves.

- Section 3 and 4 are hard to follow as its jumps between regions and different trace gas measurements. A suggestion would be to reorganize into sections focuses on specific trace gases. For example: Section 3 could just be about NO2. With the following sections.

  - 3.1: similar for 3.1.3 with summarizing Bucharest observations
  - 3.2: similar for 3.2.1 with summarizing The Jiu Valley observations
  - 3.3: Relevant discussion from Section 4 about lessons about validation

- Section 3.1.1 and 3.1.2 and their associate figures do not fit within the scope of the paper as separate sections. Any relevant discussion could fit in within the other trace gas sections, in the supplement, or omitted.

The below of the comments are organized by trace gas.
- NO2:
  - Line 209: The statement about the datasets in Figure 7 appearing consistent is not valid, which is alluded to later in the paragraph. Please reword or omit that statement in the discussion.
  - Line 221: what is the difference in time between the two measurements?
  - The statement about NO2 vmr at 300m being a proxy for NO2 VCD is not valid. It may be for that specific case but not overall. The results over the Jiu Valley even refute this statement.
  - Line 229: Is the Avantes spectrometer the Bremen Nadir instrument from Table 7? Please make descriptions consistent.
  - Not required but Figure S3 seems like a good candidate to move to the actual manuscript for comparing/contrasting with SO2. It could also be helpful to see how Figure 7 and other airborne figures translate to the TROPOMI pixels. When talking about the validation context.
  - Line 334: It seems that temporal variation could also lead to overestimation in the slope depending on how the NO2 is varying through time.
  - It should be noted that the temporal variation uncertainty quantified in this paper was specific to that area during that particular morning and more data would have to be analyzed to see if this is a typical value or not. These temporal variations are also likely much different during the time TROPOMI overpasses (not in the early morning) and on different days. Though the technique for quantifying temporal variation using the airborne data is novel and would be interesting to extend to other datasets.
  - It would also be helpful to add some more details in the writing or references about the exercise done in the first paragraph of section 4.1.3 so it can be recreated by others with similar datasets.

- H2CO and SO2
  - Line 239-240: Are there H2CO direct emissions in Bucharest? That seems to be the implication with the statement in line 239.

- This is relevant to both SO2 and H2CO since they both have the conclusion that validation of satellite H2CO and SO2 is better suited with ground-based measurements.
  - Is this a recommendation only for Romania?
  - Are there ground based measurements from AROMAT that can be discussed in terms of validation like the airborne data is? If so, add this to the discussion. If not, the conclusion that ground-based measurements would be better suited than airborne may not be valid.
  - Its mentioned that the individual flights cannot always help in validation, which is true. But systematic measurements may help as discussed in the conclusions. Please say something about this within the sections themselves.
- Emissions section (section 4.4):
  - This section lacks sufficient background on the methodology for computing fluxes and lacks the context on how this fits with the scope of the paper. Fluxes are not mentioned in the abstract, intro, nor is emission estimate validation within the requirements for validation of satellite products. Though emission estimations put into the context of satellite applications is important and evaluating that is very important scientifically from that perspective.
    - Options:
      - Omit this section.
      - Add sufficient details or references for emission flux calculations and put this into the context on how this helps with satellite product evaluation as alluded to in the latter part of section 4.3. (Though, this section with all details could potentially be a stand-alone manuscript).
        - If kept, when comparing the emissions to inventories, be sure to consider variations in emissions from hourly/daily/seasonal timescales and the AROMAT measurements were only a small subset in time.

Other comments within the text:
- Line 27-28: Veefkind et al., 2012 doesn't reference the 3.5x5.5km resolution. Refer to the switch through the Readme file or another reference that talks about it: http://www.tropomi.eu/sites/default/files/files/publicSentinel-5P-Nitrogen-Dioxide-Level-2-Product-Readme-File.pdf
- Line 45: Can it be made clear what small signals mean? Does this mean the small signal:noise ratios or more a reference to clean areas that don't have a lot of signal?
- Line 114: what are the European thresholds? Add a reference and values.
- Table 1: change GEMS to launched instead of planned.
- Throughout the paper: Spatial resolution is in km and not $km^2$. For example, 7 x $7km^2$ is not the same as 7km x 7km.
- Line 261: delete the word 'those'
- Line 396: change 'As for' to 'Similar to'
- Line 399: Start a new paragraph with the sentence starting with 'On the other hand'

---

## Referee Comment (RC2) · Anonymous Referee #2 · 13 Mar 2020

The manuscript titled The Air borne Romanian Measurements of Aerosols and Trace gases(AROMAT) campaigns at two areas provides relevance of each instrument for validation of air quality satellite (e.g., TROPOMI) products. The paper identifies a significant source of comparison error(measurement time difference), which is a useful information for the satellite validation. It summaries DL, BIAS, measurement range of several trace gas species for each instrument. However, the paper misses detailed description of instrument characteristics and measurement geometry, data used for each instrument AMF and their effects of the retrieved products. There has been no analysis about horizontal and vertical representativeness of each instrument although the campaign is to aim for validation of TROPOMI. The manuscript needs to be improved considering those major issues.

[Figure]

Abstract and Introduction: The objectives of this present study and campaign needs to be clearly distinguished. The objectives of the campaign are described in Abstract as "Their main objectives were to test recently developed air borne observation systems dedicated to air quality studies and to verify the concept of such campaigns in support of the validation of space borne atmospheric missions such as the TROPOspheric5 Monitoring Instrument (TROPOMI)/Sentinel-5 Precursor (S5P)." However, there are differences between the objectives of the campaign and those of this present work. Please address the objectives of this present study in Abstract.

Line 218: ".The comparison reveals a good agreement when averaging the forward and backward-looking Mobile-DOAS NO2 VCDs, with a MPIC/AirMAP slope of 0.93 and a correlation coefiňĄcient of 0.94." One of the campaign objectives is to identify relevance and capability of each measurement type on ground or air borne platforms for validation of TROPOMI products. There are missing of both qualitative and quantitative causes for "slope (between ground based MPIC mobile DOAS and AirMAP) of 0.93 and a correlation coefiňĄcient of 0.94". Line 224: I do not understand how "the NO2 vmr measured at 300 m a.s.l. can be used as a proxy for the NO2 VCD". Please describe how it can be used used as a proxy for the NO2 VCD. Please also use capital letter for VMR rather than vmr.

Line255-260: In comparisons between data of airborne AIRMAP, SWING, and ground based Mobile DOAS, it is important to explain if they measure the same target in terms of horizontal and vertical coverage. -If each instrument measures a target (in particular plume) at different geometry and location, there should be large differences between the retrieved NO2 VCDs. Authors need to explain reasons that cause such differences in terms of the algorithms, measurement geometries, effect of platforms, etc., in detail. -In the paper, a difference between mobile DOAS and those of airborne is partly related to air mass uncertainties. There is absence of description of NO2 AMFs for mobile DOAS and those for AirMAP and SWING. What are the input data used to calculate each AMF?

-please add schematic graph which shows instrument setup and measurement geometry (including measurement azimuth angles for target locations such as location of plume) of each instrument

Line 300: There are many sentences which mention "reference measurements". Please define "reference measurements"

Line 304: What are "typical air mass factors (AMF) used here for each species and what are the references for each AMF value for each species for each instrument?

Line 394: Please address the definition of "combined uncertainty" including how "combined uncertainty" has been calculated.

Throughout the figures tables, there no quantitative comparisons between various measurement data which were carried out at the same or similar time in the same site. Please consider adding the plots with analysis or address the reasons for not doing that.

———————————————————

---

## Author Comment (AC1) · 1 May 2020

**Reply to the comments from referee # 1**

(In red our replies, in green the text added.)

We first kindly thank the referee for his time, useful comments, and constructive criticism. We used his suggestions to prepare a new version of the manuscript.

This paper discusses the air quality measurement campaign, AROMAT, and puts it into the context of if/how these measurements can assist in future validation efforts for satellites, such as Sentinel 5P TROPOMI. The significance of this work is within the scope of AMT and is key as the air quality community works toward validating satellites that measure urban air quality (e.g., TROPOMI) and there are some novel 'take-home' messages from this work that are worthy for publishing. From what is shared in the paper, the quality of the work appears valid however there is clarity needed in some areas. This paper also needs restructuring to improve the clarity of the take-home conclusions. For example: The paper lacks details about the campaign and information about the measurements are scattered throughout the paper.

We have added two sections to describe the campaign deployment and mention more intensively the supplement when appropriate, the supplement contains more details and measurements. We also added a schematic for the overall deployment.

**Specific comments/questions:**
• The title of this paper does not clearly reflect the contents of the paper. The current title would attract readers as an overview paper for the measurements during AROMAT, but this is not the purpose of this paper nor is there a detailed overview of the entire campaigns. If agreed by the authors, I suggest changing to title to something that reflects that AROMAT could be a concept model for validation campaigns of satellite retrievals.

This paper aimed to be an overview paper but we agreed we had put too much information in the supplement so we added section 2.4 and 2.5 which describe the campaign deployments, they were previously in the Supplement. We prefer to keep our title.

• Are conclusions made about validation only valid over Romania? Or can these lessons be extended beyond Romania? Please be clear in the paper which conclusions can be extended beyond the AROMAT region.

About NO2, since Bucharest is a relatively small source compared to other cities, many other urban areas could be used as target for satellite validation using airborne measurements, we have performed other airborne measurements in Berlin and in Belgium in the AROMAPEX and BUMBA campaign which are already in the references.

About SO2 and H2CO, our conclusions are valid for Romania only since there are larger sources in other parts of the world. We have already mentioned Serbia in the conclusions.

We added a sentence to extend the scope of the conclusions for NO2:

These conclusions for NO2 above Bucharest apply to other large polluted urban areas.

• It seems that the model for the conclusions is based on TROPOMI requirements. Please comment on if/how this extends to the requirements of the other planned missions or make the specific message in the paper that the conclusions that are made are specific to TROPOMI.

The qualitative part of the conclusion is also valid for future satellites which are in Table 1 but the quantitative accuracy we give is indeed based on TROPOMI characteristics. We rephrased

Our simulations, which are based on our measurements and TROPOMI characteristics, indicate that we can constrain the accuracy of the satellite NO2 VCDs within 37 or 28%, with and without information on the aerosol and NO2 profile, respectively.

• A weakness in the general analysis is the lack of discussion on temporal variation and the time of the airborne and ground based measurements and how this relates to the time of the satellite overpass, emissions inventories, etc. The authors should keep this in mind to address through revisions.

We do not fully agree with the comment. When the time difference could explain an observed discrepancy we have mentioned it, e.g for the AirMAP to MPIC Mobile-DOAS comparisons in Bucharest

*Note that the systematic differences between AirMAP and the MPIC Mobile-DOAS at the eastern part of the ring road on 31 August 2015 were due to the time differences between both measurements.*

About the satellite validation, we have assessed the temporal error (in Section 4.1.2) for the comparison between an airborne and a satellite instrument. This is anyway more visible in this new version of the manuscript since we have included in the main manuscript the figure from the supplement which illustrates how we quantified the temporal error.

About the emissions section, we have added a sentence in the new paragraph presenting this section to emphasize the fact that we compare instantaneous emissions with yearly emission inventories:

The comparisons with reported emissions should not be overintrepeted since we compare campaign-based flux measurements performed during a few days in daytime with reported emissions which represent yearly averages. Nevertheless, they give interesting indications about the operations of the FGD units of the power plants and possible biases in emission inventories.

• Section 2 should start by painting a picture of AROMAT 1 and 2 deployments and measurements that used in this analysis. While much of this information is in the supplement and scattered throughout the paper, the general reader enters Section 3

without the proper background to assess what is being discussed. Currently, it does not effectively communicate the needed details about the AROMAT campaign before moving into the results sections. To fix this, the authors could reorganize the section by moving 2.3 to before 2.1 and 2.2. Then there needs to be discussion (and maybe a Table) that summarizes each campaign. This table and/or discussion must include time periods of each deployment, location of each deployment, payloads for the aircraft and relevant details about ground measurements (in line with Tables 4-6) and could extend into partners and other details from 2.3 as seen fit.

We thank the referee for his suggestion but we do not want to move Section 2.3 before Section 2.1 and 2.2 because we think it is clearer to present the sites before the campaign deployment on these sites. About the tables for the campaign, we also prefer to let them in the supplement since they are quite large and include ancillary measurements (ground-based in-situ, ACSM…) that we do not use later on in the main manuscript. So we think it would be distracting to put them in the main article.

But for clarity, we have moved the sections which present the practical deployments in 2014 and 2015 from the supplement to the main manuscript. We have also added a schematic for the campaign. We present this new schematics in Sect 2.3.

Figure 4 illustrates the typical instrumental deployment during the campaigns. The set-up combined airborne and ground-based measurements to sample the 3-D chemical state of the lower troposphere above polluted areas.

Table 3 does not add substantial information to this paper and that space would be more effectively be used to summarize the campaigns themselves.

We agree with the comment and moved Table 3 to the Supplement.

Section 3 and 4 are hard to follow as its jumps between regions and different trace gas measurements. A suggestion would be to reorganize into sections focuses on specific trace gases. For example: Section 3 could just be about NO2. With the following sections.
3.1: similar for 3.1.3 with summarizing Bucharest observations
3.2: similar for 3.2.1 with summarizing The Jiu Valley observations
3.3: Relevant discussion from Section 4 about lessons about validation

We thank the referee for his advices but we prefer to keep our structure because it follows in Sect. 3 a geographical order with the two sites which corresponds to the geographical presentation of Sect.2. This is interesting for a reader interested in pollution sources in Romania. In Sect. 4 we draw conclusions for each molecule, which is more interesting for a reader coming from the AQ satellite validation community. We have written a new introduction for Sect. 4.4 which presents our flux estimates.

• Section 3.1.1 and 3.1.2 and their associate figures do not fit within the scope of the paper as separate sections. Any relevant discussion could fit in within the other trace gas sections, in the supplement, or omitted.

We agree with this comment. We have moved the figure to the Supplement.

The below of the comments are organized by trace gas.

**• NO2:**

Line 209: The statement about the datasets in Figure 7 appearing consistent is not valid, which is alluded to later in the paragraph. Please reword or omit that statement in the discussion.

We meant that the measurements were consistent for each given day, not in general. We have removed this statement which was indeed misleading.

Line 221: what is the difference in time between the two measurements?

The AirMAP/Cessna VCDs correspond to several flight lines recorded between 12:00 and 13:30 UTC. We have added this info to the text, which already included the CAPS/BN-2 time.

between 12:00 and 13:30 UTC

The statement about NO2 vmr at 300m being a proxy for NO2 VCD is not valid. It may be for that specific case but not overall. The results over the Jiu Valley even refute this statement.

Although we had written at the beginning of the sentence "along this portion of the flight, which was inside the plume but outside the city," we agree that the scope of validity of this statement was not clear enough. We rephrased this paragraph for clarity:

This suggests that along this portion of the flight, which was inside the plume but outside the city, the NO2 VMR measured at 300 m a.s.l. may be used as a proxy for the NO2 VCD. Indeed, the BLH was about 1500m (Fig.S9 in the Supplement) during these observations. Assuming a constant NO2 250 VMR of 3.5 ppb in the boundary layer leads to a NO2 VCD of 1.4 x 1016 molec cm2. This estimate is close to the AirMAP NO2 VCD observed in the plume (Fig. 6). When measured at 300 m a.s.l., the NO2 VMR thus seems a good estimate of its average within the boundary layer. Note that this finding is specific to the configuration in Bucharest where we flew at 10 km from the city center and does not apply to our measurements in the exhaust plume of the Turceni power plant (Fig. 9). Future campaigns should include vertical soundings inside the Bucharest plume to further investigate its NO2 vertical distribution.

Line 229: Is the Avantes spectrometer the Bremen Nadir instrument from Table 7? Please make descriptions consistent.

Indeed, we rephrased:

from the IUP-Bremen nadir instrument

Not required but Figure S3 seems like a good candidate to move to the actual manuscript for comparing/contrasting with SO2. It could also be helpful to see how Figure 7 and other airborne figures translate to the TROPOMI pixels. When talking about the validation context.

We agree with these comments which improve the manuscript. We have moved Figure S3 to the main manuscript, merging it with the SO2 map. The figure S10 of the previous version of the Supplement, which illustrated the temporal variation, also shows how airborne measurements translate to hypothetical TROPOMI pixels. We have also moved it to the main manuscript.

Line 334: It seems that temporal variation could also lead to overestimation in the slope depending on how the NO2 is varying through time.

A systematic variation of the NO2 VCD through time would lead to a bias between reference (x) and satellite measurements (y) if the NO2 VCD, this bias could be either positive or negative.

But considering only a random variation added to the reference measurement as a noise, the dynamic range of this 'noisy' reference measurements is likely to increase, which could lead to an underestimation of the slope between x and y without this temporal noise. We have actually made some simulations to emphasize this effect and added a figure (Fig.12). Here is its legend:

Effect of an underestimation of the random error in a regression analysis simulating TROPOMI validation using airborne mapping as reference measurements of NO2 VCDs. The dynamic range (blue line) of the reference measurements increases with the applied random error. For the considered a priori random error (dashed vertical line, $1 \times 10^{15}$ molec.cm-2), this leads to an underestimation of the regression slope (red line). These simulations use the AirMAP data of 31 August 2015 (afternoon flight).

And the added text

Finally, it should be noted that these regression simulations assume a correct estimation of the temporal random error. Underestimating this error propagates in the fit of the regression slope. Figure 12 presents the possible effect of such an underestimation when the a priori random error of the reference measurements is set at $1 \times 10^{15}$ molec cm-2, using again the AirMAP observations of Fig. 5 (right panel) as input data. As the dynamic range of the reference measurements increases with the applied error, the fitted slope decreases. For a true error of $4 \times 10^{15}$, this leads for instance to an underestimation of the slope of about 5%. This effect is small but other sources of random error (e.g undersampling the satellite pixels) would add up in a real-world experiment. Wang (2017) observed such a systematic decrease of the regression slope when averaging MAX-DOAS measurements within larger time windows around the satellite overpass.

It should be noted that the temporal variation uncertainty quantified in this paper was specific to that area during that particular morning and more data would have to be analyzed to see if this is a typical value or not. These temporal variations are also

likely much different during the time TROPOMI overpasses (not in the early morning) and on different days. Though the technique for quantifying temporal variation using the airborne data is novel and would be interesting to extend to other datasets.

We agree with the comment. We had already emphasized that writing *Clearly, the NO2 VCD temporal variation depends on characteristics of a given validation experiments, such as the source locations and the wind conditions during the measurements* in Section 4.1.2 and in the conclusion *it varies with local conditions for a given experiment*.

We have further emphasized that it also depends on the time of the day and that our measurements were not at the TROPOMI overpass time.

The temporal variation also depends on the time of the day and we base our estimate here on measurements around 11:00 LT while TROPOMI overpass is at 13:30 LT.

It would also be helpful to add some more details in the writing or references about the exercise done in the first paragraph of section 4.1.3 so it can be recreated by others with similar datasets.

It seems to us that the important steps of our method are already described but we have added a reference to the section 4.1.1 describing the input data, to improve the clarity.

We simulated TROPOMI Cal/Val exercises with the spatially averaged AirMAP observations described in Sect. 4.1.1

Moreover, we have moved to the main manuscript the figure from the supplement which illustrates the quantification on the temporal error as it also shows the average of the airborne measurements at the resolution of TROPOMI so we think it improves the overall clarity of this section. This figure is described in the previous section :

Figure 11 illustrates our estimation of the temporal variation of the NO2 VCDs comparing consecutive AirMAP overpasses above Bucharest from the morning flight of 31 August 2015.

**• H2CO and SO2**

Line 239-240: Are there H2CO direct emissions in Bucharest? That seems to be the implication with the statement in line 239.

We do not know that. We clearly measured an enhancement of H2CO above Bucharest but we can not conclude on its origin. To investigate that, we would need 1) more measurements, in particular of the VOCs which are the main H2CO precursors, and 2) modelingstudies. Such work was done in particular by Johansson et al. (2014)

Johansson, J. K. E., Mellqvist, J., Samuelsson, J., Offerle, B., Moldanova, J., Rappenglück, B., … Flynn, J. (2014). Quantitative measurements and modeling of industrial formaldehyde emissions in the Greater Houston area during campaigns in 2009 and 2011. *Journal of Geophysical Research: Atmospheres, 119*(7), 4303–4322. https://doi.org/10.1002/2013JD020159

But it is outside the scope of our study. We agree that our word 'source' is misleading as it can be understood as direct source. So we have rephrased:

The difference between NO2 and H2CO spatial patterns may be explained by the different origins of NOx compared to H2CO.

This is relevant to both SO2 and H2CO since they both have the conclusion that validation of satellite H2CO and SO2 is better suited with ground-based measurements.

Is this a recommendation only for Romania?

As replied above, this is valid for Romania. There are larger SO2 sources such as volcanoes or oil industry in the Persian Gulf which may be interesting for airborne validation. For large power plants outside Romania without FGDs, one should study the satellite data to verify that their signal-to-noise ratio enables their validation with airborne measurements in practice (e.g. considering the costs). Note that the signal-to-noise may improve with geostationary platforms.

Are there ground based measurements from AROMAT that can be discussed in terms of validation like the airborne data is? If so, add this to the discussion. If not, the conclusion that ground-based measurements would be better suited than airborne may not be valid.

We have discussed the interest of our Mobile Max-DOAS (Ground based) for H2CO in Section 4.2, and the interest of the temporary SO2 cameras (Ground based) for SO2 in Section 4.3. These findings motivate the installation of automatic and static instruments since 1) MAX-DOAS are already demonstrated for H2CO VCD validation 2) TROPOMI-derived fluxes are already demonstrated and their volcanic part is already used for SO2 validation. So we consider our conclusions valid. But we agree we missed a reference for H2CO. We added a reference to Desmedt (2015) in the H2CO section, and modified a sentence:

*Indeed, long-term ground-based measurements at two sites would be useful to investigate seasonal variations of H2CO,* as already demonstrated in other sites (De Smedt, 2015).

Its mentioned that the individual flights cannot always help in validation, which is true. But systematic measurements may help as discussed in the conclusions. Please say something about this within the sections themselves.

We modified a sentence in sect 4.2:

*This limits the relevance of individual mapping flights for the validation of H2CO,* yet systematic airborne measurements would improve the statistics.

And added one in sect 4.3

As for H2CO, systematic airborne measurements would improve the statistics.

**Emissions section (section 4.4):**

• This section lacks sufficient background on the methodology for computing fluxes and lacks the context on how this fits with the scope of the paper. Fluxes are not mentioned in the abstract, intro, nor is emission estimate validation within the requirements for validation of satellite products. Though emission estimations put into the context of satellite applications is important and evaluating that is very important scientifically from that perspective.

Options:
§ Omit this section.
§ Add sufficient details or references for emission flux calculations and put this into the context on how this helps with satellite product evaluation as alluded to in the latter part of section 4.3. (Though, this section with all details could potentially be a stand-alone manuscript).
• If kept, when comparing the emissions to inventories, be sure to consider variations in emissions from hourly/daily/seasonal timescales and the AROMAT measurements were only a small subset in time.

We agree that this section was not introduced enough in the abstract and introduction and we have modified both to mention this part of the study. We added in the abstract:

We also quantify the emissions of NOx and SO2 at the two sites.

We added a sentence in the introduction:

Eventually, we use the AROMAT measurements to derive NOx and SO2 fluxes from the two sites.

Several studies already present in detail the traverse method used to quantify the fluxes with the DOAS method.e.g:

Ibrahim, O., Shaiganfar, R., Sinreich, R., Stein, T., Platt, U., and Wagner, T.: Car MAX-DOAS measurements around entire cities: quantification of $NO_x$ emissions from the cities of Mannheim and Ludwigshafen (Germany), Atmos. Meas. Tech., 3, 709–721, https://doi.org/10.5194/amt-3-709-2010, 2010.

Johansson, J. K. E., Mellqvist, J., Samuelsson, J., Offerle, B., Moldanova, J., Rappenglück, B., Lefer, B., and Flynn, J. ( 2014), Quantitative measurements and modeling of industrial formaldehyde emissions in the Greater Houston area during

campaigns in 2009 and 2011, *J. Geophys. Res. Atmos.*, 119, 4303– 4322, doi:10.1002/2013JD020159.

From AirMAP, this is also explained in the AROMAT AirMAP study

Meier, A. C., Schönhardt, A., Bösch, T., Richter, A., Seyler, A., Ruhtz, T., Constantin, D.-E., Shaiganfar, R., Wagner, T., Merlaud, A., Van Roozendael, M., Belegante, L., Nicolae, D., Georgescu, L., and Burrows, J. P.: High-resolution airborne imaging DOAS measurements of $NO_2$ above Bucharest during AROMAT, Atmos. Meas. Tech., 10, 1831–1857, https://doi.org/10.5194/amt-10-1831-2017, 2017.

We added an introduction with these references at the beginning of the section :

This section presents our estimates of the NOx and SO2 fluxes from Bucharest and the power plants in the Jiu Valley, combining our different 2014 and 2015 measurements. Campaign-based estimates of NOx emissions from large sources are relevant in a context of satellite validation since the high resolution of TROPOMI enables to derive such emissions on a daily basis Lorente(2019). Regarding SO2, as discussed in the previous section, the low signal-to-noise ratio of the satellite measurements implies averaging for several months to derive a SO2 flux (Fioletov-2019), yet campaign measurements are useful to select an interesting site and test the ground-based apparatus and algorithms.

The comparisons with reported emissions should not be overintrepreted since we compare campaign-based flux measurements performed during a few days in daytime with reported emissions which represent yearly averages. Nevertheless, they give interesting indications about the operations of the FGD units of the power plants and possible biases in emission inventories.

Our flux estimates are all based on optical remote sensing measurements. They involve integrating a transect of the plume along its spatial extent and multiplying the outcome by the plume speed, which may correspond to the stack exit velocity (camera pointing to the stack) or to the wind speed (Mobile-DOAS and imaging-DOAS). We refer the reader to previous studies for the practical implementations. Ibrahim (2010) presented the method we used for Bucharest, where we encircled the city with the Mobile-DOAS. Meier (2017) presented the AirMAP-derived flux estimations, while Johansson (2014) derived industrial emissions from a car-based Mobile-DOAS instrument as we did for the Turceni power plant. Constantin (2017) presented the fluxes based on the ULM-DOAS measurements. Regarding the SO2 Camera, we present hereafter the method and previous related works.

**Other comments within the text:**
• Line 27-28: Veefkind et al., 2012 doesn't reference the 3.5x5.5km resolution. Refer to the switch through the Readme file or another reference that talks about it: http://www.tropomi.eu/sites/default/files/files/publicSentinel-5P-Nitrogen-Dioxide-Level-2-Product-Readme-File.pdf

We agree with the comment, nevertheless Veefkind (2012) is a more complete reference for TROPOMI. So we kept it and added the readme in the references and a note after the reference to Veefkind:

the original TROPOMI resolution of 7x5.5 km2 was increased on 6 August 2019 MPC (2019))

• Line 45: Can it be made clear what small signals mean? Does this mean the small signal:noise ratios or more a reference to clean areas that don't have a lot of signal?

There are several aspects of the small signals, as Richter et al. (2013) point out:

*An additional challenge is the small signal often obtained for tropospheric species,either because their abundances are small or because it is difficult to separate the tropospheric from the stratospheric signals. In many cases, the validation measurements themselves are also not as accurate and precise for these small signals as one would like, adding the uncertainty of the validation data to that of the satellite measurement.*

We mention this paper from Richter et al. just before, L.42.:

*Richter et al. (2014) have discussed the challenges associated with the validation of tropospheric reactive gases.*

So we consider we have given the reader the useful reference to have more information on this aspect.

• Line 114: what are the European thresholds? Add a reference and values.
The reference (EEA,2019) is already given in the sentence right after, which also gives the typical value for yearly NO2 in the center of Bucharest, we added the EU limit value for yearly NO2 to compare with.

"For instance, the annual mean concentration of NO2 at the traffic stations was about 57 ug.m-3 in 2017 (EEA,2019), when the EU limit is 40 ug.m-3. "

• Table 1: change GEMS to launched instead of planned.
Corrected.

• Throughout the paper: Spatial resolution is in km and not km2. For example, 7 x 7km2 is not the same as 7km x 7km.

We do not agree with this comment, we think width x height in km2 is clear enough and shorter, thus better. This way of writing is largely used in other publications including by the TROPOMI science team. e.g. recently:

van Geffen, J., Boersma, K. F., Eskes, H., Sneep, M., ter Linden, M., Zara, M., and Veefkind, J. P.: S5P TROPOMI $NO_2$ slant column retrieval: method, stability, uncertainties and comparisons with OMI, Atmos. Meas. Tech., 13, 1315–1335, https://doi.org/10.5194/amt-13-1315-2020, 2020.

• Line 261: delete the word 'those'

Here we think our sentence is clearer as it is since we do not show all the sonde measurements, neither a random selection of them, but only the ones which detected the plume. Grammatically, we think it is also fine to use "those" … "which" since the Nobel prize in literature Bertrand Russell used this structure in his essay 'Our knowledge of the external world' (1914)

*Things are those series of aspects which obey the laws of physics.*

• Line 396: change 'As for' to 'Similar to'
Corrected.

• Line 399: Start a new paragraph with the sentence starting with 'On the other hand'

Corrected.

---

## Author Comment (AC2) · 1 May 2020

**Reply to the comments from referee # 2**

(In red our replies, in italic the text already written, in green the added text.)

We first kindly thank the referee for his time, useful comments, and constructive criticism. We used his suggestions to prepare a new version of the manuscript.

The manuscript titled The Air borne Romanian Measurements of Aerosols and Trace gases(AROMAT) campaigns at two areas provides relevance of each instrument for validation of air quality satellite (e.g., TROPOMI) products. The paper identifies a significant source of comparison error (measurement time difference), which is a useful information for the satellite validation. It summaries DL, BIAS, measurement range of several trace gas species for each instrument. However, the paper misses detailed description of instrument characteristics and measurement geometry, data used for each instrument AMF and their effects of the retrieved products. There has been no analysis about horizontal and vertical representativeness of each instrument although the campaign is to aim for validation of TROPOMI. The manuscript needs to be improved considering those major issues.

There were already two published studies (Meier et al., 2017, Merlaud et al., 2018) dedicated to the AROMAT airborne measurements, these studies include the AMF description and vertical sensitivities (box-AMF) of the airborne DOAS instruments. The Supplement already gives technical description of each instrument, giving the published references.

We agree that these two papers and the Supplement were not visible enough in the manuscript so we added several references to it (see below). We also rephrased the end of the introduction:

Two aforementioned publications focused on the AirMAP and SWING operations during the 2014 AROMAT campaign (Meier et al., 2017; Merlaud et al., 2018). In this work, we present the overall instrumental deployment during the two campaigns and analyze the relevance of these measurements for the validation of several air quality satellite products: tropospheric NO2, SO2 and H2CO VCDs.
[…]
The paper is structured as follows: Section 2 describes the two target areas and the deployment strategy. Section 3 characterizes the investigated trace gases fields in the sampled areas. Section 4 presents a critical analysis of the strengths and limitations of the campaign results while elaborating on recommendations for future validation campaigns in Romania. Eventually, we use the AROMAT measurements to derive NOx and SO2 fluxes from the two sites. The Supplement presents technical details on the instruments operated during the campaigns and presents additional information and measurements.

We also added a schematic for the geometry of the measurements.

About the horizontal representativeness, we emphasized in the conclusions that one main advantage of continuous airborne mapping is that the horizontal representativeness error cancels.

**Abstract and Introduction:** The objectives of this present study and campaign needs to be clearly distinguished. The objectives of the campaign are described in Abstract as "Their main objectives were to test recently developed air borne observation systems dedicated to air quality studies and to verify the concept of such campaigns in support of the validation of space borne atmospheric missions such as the TROPOspheric Monitoring Instrument (TROPOMI)/Sentinel-5 Precursor (S5P)." However, there are differences between the objectives of the campaign and those of this present work. Please address the objectives of this present study in Abstract.

We agree with the comment and we have added the objectives of this paper in the abstract

We present the AROMAT campaigns, focusing on the findings related to the validation of tropospheric NO2, SO2, and H2CO. We also quantify the emissions of NOx and SO2 at the two sites.

The objectives were already described in the introduction, but we rephrased it to better define the scope of the study (see above).

**Line 218:** ".The comparison reveals a good agreement when averaging the forward and backward-looking Mobile-DOAS NO2 VCDs, with a MPIC/AirMAP slope of 0.93 and a correlation coefficient of 0.94." One of the campaign objectives is to identify relevance and capability of each measurement type on ground or air borne platforms for validation of TROPOMI products. There are missing of both qualitative and quantitative causes for "slope (between ground based MPIC mobile DOAS and AirMAP) of 0.93 and a correlation coefficient of 0.94".

We are comparing collocated and almost time coincident measurements of NO2 VCDs. If everything would be perfect, the slope and correlation coefficient would both be 1. The remaining difference is small and may have several causes: instrumental bias, the small time difference of the measurements, errors of AMFs, different horizontal sensitivity.  We have added this in the text as:

The remaining discrepancy may be explained by AMFs errors and differences in time and horizontal sensitivity.

**Line 224:** I do not understand how "the NO2 vmr measured at 300 m a.s.l. can be used as a proxy for the NO2 VCD". Please describe how it can be used used as a proxy for the NO2 VCD. Please also use capital letter for VMR rather than vmr.

We had tried to give the explanation in the next sentences, which compared the NO2 VCD derived from the proxy with the AirMAP NO2 VCD measurement but we agree it was not clear enough. We have rephrased for clarity and changed vmr to VMR here and across the document

This suggests that along this portion of the flight, which was inside the plume but outside the city, the NO2 VMR measured at 300 m a.s.l. may be used as a proxy for the NO2 VCD. Indeed, the BLH was about 1500m (Fig.S9 in the Supplement) during these observations. Assuming a constant NO2 250 VMR of 3.5 ppb in the boundary layer leads to a NO2 VCD of 1.4 x 1016 molec cm2. This estimate is close to the

AirMAP NO2 VCD observed in the plume (Fig. 6). When measured at 300 m a.s.l., the NO2 VMR thus seems a good estimate of its average within the boundary layer. Note that this finding is specific to the configuration in Bucharest where we flew at 10 km from the city center and does not apply to our measurements in the exhaust plume of the Turceni power plant (Fig. 9). Future campaigns should include vertical soundings inside the Bucharest plume to further investigate its NO2 vertical distribution.

**Line255-260**: In comparisons between data of airborne AIRMAP, SWING, and ground based Mobile DOAS, it is important to explain if they measure the same target in terms of horizontal and vertical coverage. -If each instrument measures a target (in particular plume) at different geometry and location, there should be large differences between the retrieved NO2 VCDs. Authors need to explain reasons that cause such differences in terms of the algorithms, measurement geometries, effect of platforms, etc., in detail.

The instruments aim at the same target at the same time but from different locations and geometry, the ground for the Mobile-DOAS (zenith-looking) and from 3 km altitude for the airborne-DOAS (nadir-looking).

We had explained the main reasons for these differences, to our understanding, at the end of the paragraph, which also refers to our previous study which already compared airborne and mobile-DOAS measurements:

*This is partly related to air mass factor uncertainties, but probably also to 3-D effects as the plume is very thin and heterogeneous close the power plants, as discussed in Merlaud et al. (2018).*

We have to invoke another reason (3-D effects) in addition to the AMF only, since the latter can not explain the discrepancy between Mobile and airborne measurements. At the time of writing our manuscript this was still a conjectural but colleagues from another team are studying that with a 3D RT code, and their results seem consistent with what we wrote. See e.g. this presentation at ATMOS 2018

Implementation Of Three-Dimensional Box-Air-Mass-Factors In The LibRadtran Radiative Transfer Model

Schwaerzel, Marc; Emde, Claudia; Kuhlmann, Gerrit; Brunner, Dominik; Buchmann, Brigitte; Berne, Alexis

http://atmos2018.esa.int/page_session11.php

We rephrased and added a sentence to strengthen our initial explanation.

*This is partly related to air mass factor uncertainties,* but they can not explain alone such a discrepancy. Close to the power plant, the plume is very thin and heterogeneous which leads to 3-D effects in the radiative transfer, as suggested in Merlaud et al. (2018). In these conditions, the 1-D atmosphere of the radiative transfer models used to calculate the airborne AMFs may not be realistic enough and bias the VCDs measured from the aircraft.

-In the paper, a difference between mobile DOAS and those of airborne is partly related to air mass uncertainties. There is absence of description of NO2 AMFs for mobile DOAS and those for AirMAP and SWING. What are the input data used to calculate each AMF?

We agree it was not clear enough. For the airborne instruments, the NO2 profile is a box of 500 m as used in the reference we give in Meier et al. 2017 for Bucharest during AROMAT-1. We agree it was not clear enough that it was the same so we added the Phd of Andreas Meier as a reference in the AirMAP section of the supplement. This PhD includes AirMAP operation during AROMAT-2.

Note that a PhD thesis Meier (2018) describes in detail the AirMAP operations and the algorithms used to analyze the AROMAT data.

For the Mobile-DOAS, it was a zenith-only measurements and- we simply used the geometric approximation i.e. 1, as mentioned as a typical AMF value in the AMF NO2 table of the supplement. Here the AMF does not correspond to the reference (Constantin et al., 2013) since this previous work used a Chimere profile, which is not representative of the plume so close to the power plant.

We mentioned in the text and the legend of the figure that the Mobile-DOAS were zenith-only

Both AMFs actually correspond to the typical values given in the NO2 AMF table of the supplement, which we further emphasized:

Table S2 in the Supplement gives the typical AMFs used for this analysis for airborne and zenith-only Mobile-DOAS.

-please add schematic graph which shows instrument setup and measurement geometry (including measurement azimuth angles for target locations such as location of plume) of each instrument

We added a schematic (Fig.4) to explain the main campaign set-up. We did not add the azimuth however since as most of the measurements were mobile, it varied between 0 and 360°.

We have added a sentence presenting the figure in Sect 2.2

Figure 2 illustrates the typical instrumental deployment during the campaigns, which combined airborne and ground-based measurements.

This is the legend of this new figure 2 :

Geometry of the main measurements performed during the AROMAT campaigns. The Imaging-DOAS instruments map the NO2 and SO2 VCDs at 3 km altitude above

the target area while the in-situ samplers measure profiles of trace gases and aerosols. Ancillary ground measurements include Mobile-DOAS to quantify trace gases VCDs and lidars to measure the aerosol optical properties.

**Line 300:** There are many sentences which mention "reference measurements". Please define "reference measurements"

Following Richter et al. (2013), we mean "independent data with known and documented uncertainties" that we can meaningfully compare with satellite products. We have expanded the sentence in the introduction which first use this expression:

*Validation involves a statistical analysis of the differences between measurements to be validated and reference measurements,* which are independent data with known uncertainties (Von Clarmann, 2006, Richter 2013).

**Line 304:** What are "typical air mass factors (AMF) used here for each species and what are the references for each AMF value for each species for each instrument?

The sentence just after (line 305) already indicates the typical AMF: *"Table  S1 in the Supplement presents these typical AMFs and detection limits"*. The references for each AMF comes from their different reference paper. We have added that in the legend of the three AMF tables.

See the references in Sect. 2 for details on the AMF calculations of the airborne instruments. We used geometric approximations for the ground-based DOAS instruments, pointing to zenith (AMF = 1), and 22° above the hrozon (AMF = 2.7).

**Line 394:** Please address the definition of "combined uncertainty" including how "combined uncertainty" has been calculated.

We had already explained this definition at the beginning of the sentence*: Adding in quadrature the biases of the SO2 VCDs for airborne measurements (40%, Table 6) and for TROPOMI (30%, Table 2)  already leads to a combined uncertainty of …' It seems already clear to us.*

Throughout the figures tables, there no quantitative comparisons between various measurement data which were carried out at the same or similar time in the same site. Please consider adding the plots with analysis or address the reasons for not doing that.

We had put the quantitative comparisons in the Supplement, where there are SWING vs AirMAP comparisons for NO2 and SO2 (Fig. S1 and S2) and AirMAP vs Mobile-DOAS measurements (S6), with correlation coefficients, slopes, intercepts, and number of points.  We also show a CAPS versus NO2 sonde intercomparison (Fig. S5) but here it did not make sense to quantify the slope as the sonde was calibrated with the CAPS data. So we think we have already the quantitative intercomparisons needed to support the conclusions of our study. But we agree it was not visible enough in the main article so we added references to these figures in

the section 3.2.2 and 4.1.2 where these corresponding measurements are discussed and used:

In Sect. 3.2.2, we had already written that the SWING and AirMAP VCDs agree within 10%, we have added:

Figure S4 in the Supplement shows the corresponding time series of SWING and AirMAP SO2 DSCDs.

In Sect. 4.1.2

Figure S3 in the Supplement presents the corresponding AirMAP and SWING NO2 DSCDs.

Moroever, we added a quantitative description in the conclusion for the comparisons between airborne and mobile-DOAS in Bucharest.

*These measurements agree* within 7\% *with ground-based measurements*

---

## Author Response (AR2)

**Response to the review**

We thank the editor and referee for their time and the accurate and constructive remarks of this second review. We have used their comments to improve the text.

In red our reply, in green the text added.

**Comments from the editor**

Dear authors,
Your revised manuscript has improved after taking into account the comments and suggestions by two referees. Ref#1 still has a number of useful suggestions that need to be implemented to make the paper fit for publication in AMT.
Please follow up on the constructive suggestions and comments from Ref#1, and then submit a revised version of your manuscript. Specifically:
- trim the supplement to only contain information relevant to the analysis done in the manuscript

We removed the figure with the PICARRO soundings and of the in-situ measurements of aerosol chemical compositions, NO2 and SO2 in Turceni, which we did not use in the analysis. We also removed Section S2.6
- reconsider the colour schemes in the map figures

We have corrected that.
- reconsider the title

We changed the title.
- acknowledge that the 4e+15 estimate is indeed fairly particular to the Romanian campaign and day sampled

We have removed this estimate from the abstract and emphasized its context in the conclusion.

**Comments from referee 1**
Second review of 'The Airborne Romanian measurements of Aerosols and Trace gases (AROMAT) campaigns'. One very positive change I did notice and that I think the flow improved a lot through the removal of the sections 3.1.1 and 3.1.2 in the original submission. I still have some concerns though paper that needs revisions before publishing and they are mostly concerns that were brought up in my original review.

This title still does not reflect the significance of this paper. I strongly suggest to the authors to add to the title to reflect the attempts for assessing validation or campaign strategies for current/future missions satellite missions. It is possible to have an all-encompassing title that advertises AROMAT and the focus on take-home messages in the paper. Possible suggestion:
'Conceptual satellite validation strategy assessments during the AROMAT campaigns'
I only stress this again because it will help the paper reach a broader audience. This paper's purpose isn't just about advertising a campaign.

We agree with the referee and the editor. We changed the title to

**Satellite validation strategy assessments based on the AROMAT campaigns**

Another issue I have is the strength of conclusions drawn on the temporal variation quantification of 4E15. It is a novel way to do this calculation, but the conclusions drawn from one morning of data over 8 'satellite' pixels should be lightly taken. This one number could be very atypical, but we don't know since it is just one example. This is somewhat stated in the manuscript but not covered when assessed in the abstract/conclusions.

One dependency I didn't see other than the mention of meteorology, emissions, time of day, is that that temporal variation is also dependent on the time window asessed which can changed (e.g., this example is about an hour time window but that window could be smaller in another experiment, which will also be under different meteorology, emissions, etc.). So the recommendation is to draw back the focus on that one number of 4e15 in your conclusions.

Edits needed at the very least: In the abstract and conclusions if you are going to state this number of 4e15, you should also state that this is quantified from one sample under one set of conditions in the morning timeframe and how this can vary in time of day and meteorology in the abstract and conclusions as well (I know it is already stated in the manuscript). Also, there is not enough information to say this is typical for AROMAT with only one example, so the added information would be in place of rewording of this phrase. I think an important conclusion that this study brings to light is that the satellite air quality community should further investigate of the impact of temporal variation on results. There are datasets out there where this can be explored more conceptually than just this one example, so give that to readers to go out and explore further.

We agree with the comment. We have removed the 4e15 from the abstract. In the conclusion, we kept the quantitative estimate of 4e15 but further stressed that this number should not be overinterpreted and repeated the finding that neglecting the random part of this time error could low bias the regression slope since this may be more transposable.

In the abstract we rephrased to:

However, we show that the temporal variation of the NO2 VCDs during a flight might be a significant source of comparison error.

And in the conclusion:

For a single morning flight above Bucharest, we have estimated the random part of this temporal error to be about 4 x 10$^{15}$ molec cm$^{-2}$. In the AROMAT conditions, underestimating this error would lead to a low bias in the regression slope between satellite and airborne measurements. This temporal error varies with local conditions for a given experiment but the satellite air quality community should further investigate this effect.

Many times in this manuscript it is said that measurements are time and space coincident, but it is never said what the time/space constraints are. Convince me (the audience) they are coincident by listing these constraints. Other times it says measurements are not time coincident, but they are still compared and saying they are different because of time difference but a reader has no idea what that time difference is. This is really critical for interpreting the data. Here are a few examples…I am not sure if this list is all inclusive:

• Lines 247-251. If there are time coincident measurements in the morning, then why aren't those shown to compare the datasets? Or the data that isn't temporally coincident shouldn't be shown at all. At the very least the time difference should be quantified.

We agree with the comment in general. However, Figure 5 illustrates the general deployment on 31 August 2015 and we want to keep it as it is. We compare more accurately AirMAP and the CAPS data in Figure 6, for which we already gave the time windows of the two measurements:

(L.255-256: *between 12:30 and 12:55 UTC, while AirMAP onboard the Cessna was mapping the city between 12:00 and 13:30 UTC))*

We also quantitatively compare the AirMAP to the MPIC Mobile-DOAS on Fig. S2. We had written that these measurements were *more simultaneous* than in Fig 5. This was indeed not accurate enough and we have added the criteria at the beginning of the sentence:

Considering only collocated measurements with a maximum time difference of 45 minutes…

We also add the flight time and collocation information in the caption of Fig. S2.

• Lines 293-295 and Lines 323-327: The aircraft and mobile measurements are said to be simultaneous, but I don't see data or text convincing of that as the graph is just by longitude. Plume structures can change very quickly in time, so if the time difference that may seem small could still be an impact (even if it's as little as a half hour or so) this could explain most of the mismatch as well. Unless they are coincident down to just a few minutes then I see reason to question the temporal effects.

The Mobile-DOAS was sampling under the plume while the aircraft was mapping the area. The aircraft measurements correspond to portions of 3 flight lines between 09:54 UTC and 10:17 UTC. For the figure, we extracted the Mobile-DOAS and airborne measurements along the road in this time window, so the maximum time difference is 23 min. Indeed the plume structure may quickly vary but we also got this kind of comparisons during AROMAT-1, when we had several trips with the car under the plume, close in time to the aircraft overpass. Moreover, there is a theoretical explanation for such discrepancies invoking 3D effects of the radiative transfer as we already discuss in the text. We added the time information.

The airborne data correspond to three portions of flight lines recorded between 09:54 and 10:17 UTC. The BIRA Mobile-DOAS instrument was sampling the plume during this time so the maximum time difference is 23 minutes.

And in the caption of figure S5.

Both airborne and ground-based data were recorded between 09:54 and 10:17 UTC.

• Overall statement: All flights and ground based datasets shown in the manuscript should have noted time windows for data collection.

We agree with the comment. We added the time information where we found it was missing, as in the H2CO figure description.

And in the captions of figure 6, 7, 8, and 13.

Presenting Fig. 7 and 13 in the manuscript, we added the time info as well.

Figure 7 […] The airborne data shown correspond to the second overpass (07:46-08:23 UTC) while the Mobile-DOAS were recorded between 08:13 and 10:00 UTC.

Figure 8 presents the horizontal distribution of the $NO_2$ VCDs in the Jiu Valley measured with the MPIC Mobile-DOAS on 23 August 2015 between 08:07 and 14:16 UTC.

Figure 13 presents a scatter plot of the slant columns of $NO_2$ and $SO_2$ for the ultralight flight of 26 August 2015, which detected the four exhaust plumes of the Valley between 08:31 and 11:04 UTC.

Minor edits:
• Line 53, add the year to the Richter reference.
Added.
• Line 183: Where is RADO? Should this be in the map in Figure 2? Or does it match up to one of the existing labels in Figure 2?

RADO is the INOE atmospheric observatory in Magurele, which is already in Fig 2. We clarified the location in the text.
• Line 218: Specify which nadir-looking spectrometer was on the UGAL ultralight.

We changed the wording to 'the ULM-DOAS instrument', which is presented in the instrument list of the Supplement.
• Line 216: Splitted should be split

Corrected
• Line 294: Which mobile DOAS instrument is this? Additionally, I get the feeling when I read this

paper through a few times that there may be other instances where mobile DOAS is stated but not identifying which one. Check these types of details.

We have specified the instrument (BIRA). We have checked the other occurrences of Mobile-DOAS and completed when needed (section 4.4.1 and caption of Table 7).
• Line 565: I missed where it was quantified that the no2 ground and airborne measurements agree with 7%. Can you clarify? Is it referring to the slope between MPIC/AirMAP from the supplement? If so, it is only between the two instruments. The sentence in the conclusion is broad making it seem like all NO2 measurements are within that agreement.

Indeed, it is the aforementioned slope. We thought the scope was clear since we used 'These', which refers to the previous sentence giving the scope. But we agree it could be misleading, we rephrased:

In the AROMAT conditions, airborne measurements were consistent with ground-based observations within 7\%...
• Line 579-581: The 'structure' was not shown for HCHO to be able to draw whether it is visible from daily satellite overpasses. Please rephrase.

We rephrased to .

Due to the lower signal to noise ratio of the $H_2CO$ observations, it is difficult to use such daily measurements for satellite validation.

• Figure 7 and Table 3: IUP-UB nadir only compact spectrometer is not listed in Table 3 for NO2, but the caption says it's from the IUP-UB for both NO2 and HCHO. I also saw some inconsistencies in IUP-UB vs IUP-Bremen in terms of naming convention, too.

We have changed IUP-UB to IUP-Bremen for consistency, and we added this instrument in Table 3.
• Figure 9: Is the last two digits in the legend correspond to hour? Is it in UTC? Please be clear in the caption. Another suggestion that will make this figure infinitely easier to investigate would be to make the y-axis only expand up to 2000m or so. There is not really anything changing above 1.5 km.

Indeed. We clarified the caption. However, we prefer to keep the y-axis as it is as it clearly points out that the upper troposphere is completely free of NO2.

• Figure 11: the color bar in the top left figure is different from the rest of the figure. Additionally, you mention Fig. S9 in the caption but looking at Fig. S9, I do not see 3 mobile DOAS sites.

We thank the referee for pointing us an error in this figure. We have corrected it.

• Figure S3 and S4: The colors for the lines in the caption don't appear to be right. The fit lines I see appear to be yellow (not blue) are very thin and hard to see.

We have updated the figure to make the straight lines thicker and corrected the color in the caption.

• Figure S11: add in the caption where these in situ measurements are collected in that domain. Are they at the Turceni powerplant? And are they at the ground?

Following the remark of the editor, we removed this figure and others to trim the supplement, as we did not use them in the analysis.

• Sections S2.6 doesn't have locations listed for these in situ measurements. And that section doesn't have SO2 or NO2, though Figure S11 shows NO2 and SO2 in situ measurements.

We removed Sect S2.6 (same reason as previous reply)

**Relevant changes**

- New title

- Removed 4e15 for the temporal variation from the abstract and stress that it corresponds to a single flight in the conclusion

- Removed the material in the supplement which was not used in the analysis

- Corrected Fig 11 where some of the color bars were wrong

- Added the time information on the measurements where it was missing

[revised manuscript text omitted]